# Clinical Performance and Retention of Partial Implant Restorations Cemented with Fuji Plus^®^ and DentoTemp™: A Retrospective Clinical Study with Mechanical Validation

**DOI:** 10.3390/medicina61122183

**Published:** 2025-12-08

**Authors:** Sergiu-Manuel Antonie, Laura-Cristina Rusu, Ioan-Achim Borsanu, Remus Christian Bratu, Emanuel-Adrian Bratu

**Affiliations:** 1Clinic of Implant Supported Restorations, “Victor Babes” University of Medicine and Pharmacy Timisoara, 2 Eftimie Murgu Sq., 300041 Timisoara, Romania; antonie.sergiu@umft.ro (S.-M.A.); remusbratuchristian@gmail.com (R.C.B.); ebratu@umft.ro (E.-A.B.); 2Multidisciplinary Center for Research, Evaluation, Diagnosis and Therapies in Oral Medicine, “Victor Babes” University of Medicine and Pharmacy Timisoara, 300041 Timisoara, Romania; 3Clinic of Oral Pathology, “Victor Babes” University of Medicine and Pharmacy Timisoara, 300041 Timisoara, Romania

**Keywords:** dental prosthesis, implant-supported, dental cements, zirconia, glass ionomer cements, retrospective studies

## Abstract

*Background and Objectives*: Cement-retained implant restorations are widely used because they offer favorable esthetics and a passive fit. Their long-term performance is strongly influenced by cement selection and surface conditioning. This study compared the clinical performance of a resin-modified glass ionomer cement (Fuji Plus^®^) with a provisional acrylic-urethane cement (DentoTemp™) in partial implant restorations. *Materials and Methods*: A retrospective cohort of 40 patients with three-unit implant-supported fixed dental prostheses was followed for at least three years. Restorations were fabricated from zirconia or metal-ceramic frameworks and cemented with either Fuji Plus^®^ or DentoTemp™. Clinical outcomes included retention, failure events, marginal adaptation, and peri-implant tissue response. In order to illustrate the impact of cement type and abutment height, mechanical testing was also carried out on standardized in vitro models; however, these tests were not powered for formal hypothesis testing. Although OCT images were included in this study only as illustrative examples from our clinical database and were not obtained from the analyzed cohort, OCT may be a useful tool for non-invasive assessment of marginal fit. *Results*: Zirconia restorations showed a retention rate of 95 percent, while metal-ceramic restorations reached 85 percent. All four failures occurred in cases cemented with DentoTemp™, giving an overall retention rate of 80 percent for this group. Fuji Plus^®^ achieved complete retention in all cases. Re-cementation with Fuji Plus^®^ successfully resolved the failures. Marginal adaptation was evaluated qualitatively because radiographic analysis did not enable accurate measurement of marginal gaps. When cement remnants were found, mild peri-implant inflammation was seen. *Conclusions*: Within the limitations of this small retrospective, non-randomized cohort, Fuji Plus^®^ demonstrated a tendency toward better peri-implant tissue response and longer-term retention than DentoTemp™. These findings should be interpreted as preliminary and exploratory rather than conclusive. Fuji Plus^®^ may be a suitable option for definitive cementation in partial implant restorations, while DentoTemp™ may be considered in selected situations where retrievability is important.

## 1. Introduction

Implant-supported fixed dental prostheses (FDPs) have become the standard of care for partially edentulous patients, offering high long-term survival rates and favorable functional and esthetic outcomes. Cement-retained prostheses are frequently used due to their ability to achieve passive fit, esthetic integration, and simplified fabrication compared to screw-retained alternatives [1,2,3,4]. Fuji Plus^®^ is a resin-modified glass ionomer (RMGI) cement, which differs fundamentally from 10-MDP–containing resin cements (adhesive resins) and self-adhesive resin cements. RMGIs exhibit an acid–base reaction reinforced by resin components, providing chemical adhesion to tooth structure and certain metals, together with moderate moisture tolerance but limited retrievability. In contrast, 10-MDP–based resin cements depend on micromechanical retention and functional monomer bonding to substrates such as zirconia, achieving high bond strengths but requiring strict isolation.

In our clinical practice, DentoTemp™ (an acrylic-urethane provisional cement) has repeatedly shown stable medium- to long-term performance, often comparable to definitive cements in partial implant restorations. Importantly, DentoTemp™ was intentionally selected as a definitive cement only in cases where retrievability was clinically desirable, which inherently introduces clinician-driven selection bias. As a result, the two cement groups were not equivalent at baseline, reflecting real-world clinical decision-making rather than a controlled allocation. This study was therefore designed to formally compare Fuji Plus^®^ and DentoTemp™ as two cements frequently encountered in real-world implant prosthodontics, despite their different classifications.

The clinical performance of cement-retained prostheses is strongly influenced by the choice of luting agent. Resin-based cements, particularly those containing functional monomers such as 10-methacryloyloxydecyl dihydrogen phosphate (10-MDP), provide excellent mechanical retention [5,6,7]. However, residual excess cement may trigger biological complications if not fully removed [8,9,10,11]. Self-adhesive resin cements, while more user-friendly, have shown lower bond strength and higher debonding rates in implant prosthodontics [12,13,14].

In addition to cement type, marginal adaptation is a critical determinant of long-term success, as discrepancies may predispose to peri-implant inflammation and mechanical failure [15]. Initial evaluation of prosthetic fit is performed by direct visual inspection and tactile assessment. Conventional radiography is widely used, but its limited resolution (≥75–150 μm) prevents reliable detection of microgaps or cement remnants [16,17,18]. Optical coherence tomography (OCT), a high-resolution and non-ionizing modality, has emerged as a potential tool for assessing marginal fit with greater precision [19,20,21]; however, in the present study, it served solely an illustrative role. OCT images included in this manuscript originate from external clinical cases and were not obtained from the analyzed cohort.

While numerous in vitro studies have investigated bonding strategies and luting agents, there is limited clinical evidence directly comparing definitive cements with materials often used provisionally but occasionally applied in practice as definitive agents. In particular, the use of provisional cements such as acrylic-urethane-based materials remains underexplored, despite their real-world use in selected cases where retrievability is desirable.

Therefore, the objective of this retrospective clinical study was to compare the retention, failure patterns, marginal adaptation, and biological response of partial implant-supported restorations, fabricated from zirconia or metal-ceramic frameworks, cemented with either a resin-modified glass ionomer (Fuji Plus^®^, GC Corp., Tokyo, Japan) or an acrylic-urethane provisional cement (DentoTemp™, Itena Clinical, Villepinte, France). While Fuji Plus^®^ is expected to provide superior long-term retention as a definitive cement, because its chemical composition and design imply that, our clinical experience indicates that DentoTemp™ can also perform reliably as a definitive luting agent in selected scenarios where controlled retrievability is desirable.

## 2. Materials and Methods

### 2.1. Study Design and Ethics

This retrospective cohort followed the Declaration of Helsinki. Approval was obtained from the Ethics Committee of Victor Babeș University of Medicine and Pharmacy Timișoara (approval no. 16/21 January 2025), and all patients signed written informed consent for scientific use of anonymized data. Because treatments were completed before analysis, the study was not prospectively registered; this is acknowledged as a limitation.

The study included adult patients who received cement-retained partial implant restorations supported by two dental implants. A minimum clinical follow-up of three years was required for inclusion, and all cases had complete documentation, including radiographs and clinical records. Because this was a retrospective analysis of completed treatments, the study was not prospectively registered in a clinical trial registry.

In addition to the clinical evaluation, an exploratory in vitro component was conducted to illustrate the influence of cement type and abutment height under controlled conditions (see Section 2.3). Four representative three-unit models were prepared, corresponding to 5 mm and 7 mm abutment heights, and used to fabricate both zirconia and metal–ceramic restorations. Each restoration was tested with both DentoTemp™ (Itena Clinical, Villepinte, France) and Fuji Plus^®^ (GC Corp., Tokyo, Japan) [22,23]. This laboratory setup was intended as an illustrative complement to the clinical findings rather than a stand-alone in vitro trial.

In Figure 1 are presented two cases, one cemented with Fuji Plus and one with DentoTemp™.

We acknowledge that this study compares a provisional cement (DentoTemp™) with a definitive cement (Fuji Plus^®^). In our clinical protocol, DentoTemp™ was intentionally selected as a definitive material only in situations where retrievability was clinically desirable. This clinician-driven allocation introduces inherent selection bias, and therefore the two cement groups were not equivalent at baseline. These real-world indications, rather than random assignment, guided cement choice, and this limitation is explicitly recognized in the interpretation of the results.

### 2.2. Patient Selection and Grouping

A total of 40 patients were selected for inclusion based on strict eligibility criteria. Patients were required to have complete clinical and radiographic documentation, including orthopantomography (OPG), cone beam computed tomography (CBCT), intraoral photographs, and follow-up records. Implants from Bredent, MIS, MegaGen, and Straumann systems were included. Definitive abutments were torqued following the standard protocol recommended by each manufacturer, and all restorations were delivered as cement-retained partial prostheses supported by 2–3 implants, with spans ranging from 2 to 4 elements. Non-smokers and light smokers (≤5 cigarettes/day) were eligible, while heavy smokers were excluded. Exclusion criteria included uncontrolled systemic diseases (e.g., diabetes, osteoporosis), parafunctional habits (e.g., bruxism), progressive peri-implant bone loss at baseline, screw-retained restorations, and non-standardized cementation protocols. Standardized cementation protocol (required for inclusion) was defined as: isolation with cotton rolls/high-volume suction (rubber dam where feasible), abutment decontamination with chlorhexidine, substrate-specific surface pretreatment (Section 2.4.1), controlled cement application, and sequential excess removal (microbrush at margins → floss/interprox brushes interproximally → targeted ultrasonic tip at emergence profile), followed by immediate radiographic/clinical verification.

Patients were included in the study and equally distributed across restoration material types: 20 patients received zirconia restorations, and 20 patients received metal-ceramic restorations.

Each material group was further divided based on the cement type used for final cementation: 10 zirconia restorations were cemented with DentoTemp™, and 10 with Fuji Plus^®^; 10 metal-ceramic restorations were cemented with DentoTemp™, and 10 with Fuji Plus^®^.

This resulted in a balanced four-arm structure:Group 1: Zirconia + DentoTemp™;Group 2: Zirconia + Fuji Plus;Group 3: Metal-Ceramic + DentoTemp™;Group 4: Metal-Ceramic + Fuji Plus.

Abutment height (5 mm or 7 mm) was selected clinically according to soft tissue thickness and crown height space. In this retrospective analysis, abutment height was documented for all failed cases, allowing descriptive evaluation of the relationship between failures and abutment height. Given the limited sample size and low number of events, no formal multivariate analysis was performed.

Without using formal multivariate modeling, this grouping design allowed for an organized descriptive comparison of restorative material, cement type, and abutment height with regard to retention, failure rates, and soft tissue response.

### 2.3. Mechanical Model Validation

To complement the retrospective clinical findings, an exploratory mechanical validation was conducted to assess the influence of cement type and abutment height on the retention of partial implant restorations under standardized laboratory conditions. These tests were designed solely as an exploratory and illustrative comparison to contextualize the clinical findings, not as a fully powered validation study, and no statistical inference was attempted.

Two representative three-unit fixed dental prosthesis (FDP) models were fabricated to replicate the most common clinical configuration, two implants supporting a posterior bridge. Each model was mounted on titanium implant analogs matching the clinical implant systems. One FDP was manufactured in monolithic zirconia and the other in a metal framework suitable for metal–ceramic veneering. For each material, two abutment heights (5 mm and 7 mm) were prepared, yielding four test assemblies (zirconia–5 mm, zirconia–7 mm, metal–ceramic–5 mm, metal–ceramic–7 mm). Figure 2 illustrates the abutments and corresponding restorations.

Each restoration was cemented, tested, cleaned, and re-cemented with the alternative cement so that both Fuji Plus^®^ and DentoTemp™ were evaluated for every configuration. Seating pressure and working/setting times followed the manufacturer’s instructions. Although intaglio surfaces were refreshed between cycles, repeated cementation may induce minor surface alterations; thus, the results should be interpreted as exploratory.

Debonding was performed under uniaxial tensile loading using a universal testing machine (Department of Mechanics) at a crosshead speed of 0.5 mm/min, consistent with established implant-cement retention protocols. A custom jig ensured axial load application. Failure was defined as the first abrupt drop in force corresponding to restoration dislodgement, and peak pull-off force (N) was recorded.

For each combination of restorative material, abutment height, and cement type, *n* = 10 independent measurements were obtained, resulting in 80 total tests. Between recementations, cement residues were removed using a standardized protocol consisting of plastic-instrument debridement, ultrasonic cleaning in isopropyl alcohol (5 min), gentle air-abrasion with 50 µm Al_2_O_3_ at ≤1.0 bar, steam cleaning, and air-drying.

This pull-off protocol follows recognized methods for assessing the retentive strength of cemented implant restorations [24,25,26,27,28] and provides a standardized framework to illustrate the relative influence of cement type and abutment height under controlled conditions, without claiming definitive mechanical validation.

### 2.4. Surgical and Prosthetic Procedures

All procedures were performed by three experienced clinicians (Prof. Dr. Bratu, Dr. Antonie, and Dr. Borșanu) who were directly responsible for both the surgical and prosthetic phases. They were assisted only by auxiliary staff, including nurses and radiology personnel, to ensure standardized workflow and procedural consistency. Prior to implant placement, each patient underwent a thorough clinical and radiological evaluation, including OPG and CBCT. These diagnostic tools were used to assess bone volume, proximity to anatomical structures, and soft tissue conditions, enabling prosthetically driven planning.

Following radiological and clinical assessment, at least two dental implants were placed per patient under sterile conditions, with attention to achieving primary stability and ideal prosthetic positioning. Implant site preparation and insertion were performed in accordance with the manufacturer’s guidelines. Healing abutments were placed at surgery, and patients were instructed on post-operative hygiene and follow-up.

A healing period of three to six months was allowed to ensure osseointegration, during which patients were periodically re-evaluated. At the end of the healing phase, provisional restorations were fabricated and placed to guide soft tissue contouring and occlusal adaptation. The provisional phase also allowed clinicians to assess abutment height requirements, functional loading, and interocclusal space.

Definitive restorations were delivered approximately one year after implant placement. Final abutments with a height of either 5 mm or 7 mm were selected according to soft tissue thickness and crown height space. Restorations consisted of three-unit cement-retained fixed dental prostheses, either fabricated from monolithic zirconia or metal-ceramic materials.

#### 2.4.1. Surface Preparation

Surface preparation protocols were applied based on the restorative material:Zirconia restorations were treated with airborne-particle abrasion using 50 μm aluminum oxide at 1.5–2.5 bar pressure, followed by application of a 10-MDP-containing primer (Clearfil Ceramic Primer Plus, Kuraray Noritake, Tokyo, Japan) to enhance chemical bonding [6,24,29].Metal-ceramic restorations underwent mechanical abrasion of the metal framework, followed by application of a metal primer (Alloy Primer, Kuraray Noritake, Tokyo, Japan) or silane coupling agent (Monobond Plus, Ivoclar Vivadent, Schaan, Liechtenstein), depending on the presence of a ceramic veneering layer [25,30].

Restorations were ultrasonically cleaned in isopropyl alcohol or ethanol for five minutes, air-dried, and stored in a clean container prior to cementation. In Figure 3, photos of these two types of restorations are shown, from different angles, in order to see the characteristics of both surfaces.

#### 2.4.2. Cementation

The abutments were cleaned with chlorhexidine and dried before cement application. Two luting agents were used: DentoTemp™ (Itena Clinical, Villepinte, France), classified as a provisional acrylic-urethane cement, and Fuji Plus^®^ (GC Corporation, Tokyo, Japan), a resin-modified glass ionomer. Although DentoTemp™ is categorized as a temporary cement, in our clinical protocol it was also applied as a definitive luting agent in selected cases. The possibility of occasional decementation was considered acceptable, since it allowed retrievability without significant clinical complications. Fuji Plus^®^ was used as a definitive luting agent where long-term stability was prioritized. Informed consent explicitly disclosed that DentoTemp™, though used as a definitive luting agent in selected cases, carries a higher risk of decementation but offers superior retrievability. Isolation was performed using cotton rolls, high-volume evacuation, and retraction cord when indicated. According to the manufacturer’s instructions, DentoTemp™ provides a working time of 45–60 s, with an initial set between 2 min and 2 min 30 s at 23 °C. Fuji Plus^®^ is a resin-modified glass ionomer (RMGI) definitive cement; its manufacturer specifies a working time of 2 min at 23 °C (with extended-working-time versions up to 3 min 30 s), and start of finishing at ~4–5 min. Cementation followed these timeframes strictly.

After seating, excess cement was carefully removed using ultrasonic scalers, dental floss, and interdental brushes, ensuring that no remnants remained at the peri-implant sulcus. Post-cementation occlusal adjustments were performed, and each patient was enrolled in a recall protocol for ongoing evaluation.

### 2.5. Outcome Measures and Statistics

The clinical performance of the cement-retained partial implant restorations was assessed through predefined primary and secondary outcome measures, focusing on retention, failure events, marginal adaptation, and soft tissue response.

Primary Outcome Measures:Retention Rate: Defined as the proportion of restorations that remained functionally intact without re-cementation during the three-year follow-up period.Failure Rate: Defined as any restoration requiring re-cementation due to loss of retention, without mechanical fracture or abutment damage.

Secondary Outcome Measures:Marginal Adaptation: Evaluated radiographically based on periapical and panoramic images. Although precise quantification of marginal discrepancies under 150 μm was not feasible, qualitative comparison of restoration-to-abutment fit was performed. No quantitative or semi-quantitative radiographic measurements were used in this study because radiographs cannot have resolution below 75–150 µm. OCT was introduced as an alternative method to illustrate the marginal gap. It can overcome the limitations of radiographs and has the potential for high-resolution imaging in future studies about cementation assessments [19].Biological Response: Clinical examination of peri-implant tissues was conducted to detect signs of inflammation, bleeding on probing, or suppuration. The presence of residual subgingival cement was recorded and correlated with soft tissue outcomes.

All collected data were recorded and analyzed using Excel and the online platform StatsKingdom [31]. Categorical variables (e.g., retention and failure events by cement type and restorative material) were compared using Fisher’s exact test, given the small subgroup sizes. Continuous variables such as age were assessed using independent-samples *t*-tests. Kaplan–Meier survival analysis was performed to evaluate time to failure. A significance threshold of *p* < 0.05 was adopted. Marginal adaptation was therefore interpreted qualitatively, based on the presence or absence of gross discrepancies, overhanging contours, or obvious gaps, rather than by attempting to derive absolute gap values.

## 3. Results

### 3.1. Patient Demographics and Clinical Grouping

A total of 40 patients were included in the study and evenly distributed across four clinical subgroups according to restoration material (zirconia vs. metal-ceramic) and cement type (DentoTemp™ vs. Fuji Plus). The mean age across all groups ranged from 45.0 to 51.1 years, with no statistically significant difference in age distribution. The gender ratio was approximately balanced, with a slight predominance of male patients in the zirconia + DentoTemp™ and metal-ceramic + Fuji Plus subgroups.

Maxillary and mandibular implant locations were represented in all groups. Zirconia + Fuji Plus restorations showed a higher number of maxillary placements (7 out of 10 cases), while the distribution was nearly even in the other subgroups. Each patient received a three-unit cement-retained partial restoration supported by at least two implants. In Table 1, all the demographic data is gathered and grouped by the specifics of each clinical case.

### 3.2. Mechanical Validation Outcomes

Mechanical testing confirmed differences in retention between the two cements and highlighted the role of abutment height. Restorations cemented with Fuji Plus^®^ demonstrated the highest resistance, with peak loads reaching ~294 N for 7 mm abutments and ~236 N for 5 mm abutments. In contrast, DentoTemp™ showed substantially lower resistance, peaking at ~143 N for 7 mm abutments and ~89 N for 5 mm abutments. The 7 mm abutments consistently provided higher retention than 5 mm abutments, regardless of cement type. These findings (Figure 4) illustrate that both cement choice and abutment height may influence retentive strength under uniaxial tensile loading, with higher values consistently observed for Fuji Plus^®^ and for 7 mm abutments in this exploratory setup.

### 3.3. Retention Rates and Failure Events

#### 3.3.1. Zirconia vs. Metal-Ceramic

Retention success varied according to the restorative material. Zirconia-based restorations exhibited a 95% retention rate, with only one failure over the three-year follow-up period. In contrast, metal-ceramic restorations had an 85% retention rate, with three recorded failures attributed to loss of retention. This difference did not reach statistical significance (Fisher’s exact test, *p* = 0.60), but retention was numerically higher in zirconia. These findings suggest that zirconia restorations, when properly treated and bonded, achieved slightly higher long-term stability under the tested conditions. This information is also presented in Table 2. Figure 5 was designed as a stacked bar chart illustrating the proportion of successful and failed restorations for each material group, complementing the numerical data in Table 2. This dual presentation highlights both absolute numbers and relative failure percentages for visual clarity.

#### 3.3.2. DentoTemp™ vs. Fuji Plus

All four failures occurred in restorations cemented with DentoTemp™. No debonding was recorded in restorations cemented with Fuji Plus^®^, which resulted in a 100% retention rate across both zirconia and metal-ceramic groups. Although Fisher’s exact test did not demonstrate a statistically significant difference (*p* = 0.114), this value is underpowered given the small subgroup size (*n* = 10 per arm) and should therefore be interpreted only as an exploratory descriptive trend rather than a meaningful statistical finding. These findings reinforce the critical influence of cement type on the long-term stability of implant-supported restorations and are presented in Table 3.

Re-cementation with Fuji Plus^®^ was performed in all failed cases as part of our standard clinical workflow. These cases were not re-entered into the survival analysis to avoid influencing comparative outcomes. After re-cementation, no further loss of retention was observed during follow-up.

#### 3.3.3. Time to Failure

In Table 4 are gathered all the information about time-to-failure of dental restorations. Among the four restorations that required re-cementation, three failures occurred between 7 and 15 months post-cementation. The only zirconia failure occurred at 12 months and had initially been cemented with DentoTemp™. The three metal-ceramic failures occurred earlier (7, 10, and 15 months), also with DentoTemp™.

All re-cementations were performed using Fuji Plus, in accordance with our standard protocol for managing definitive cement debonding, after which no additional failures were recorded. As noted above, these re-cemented restorations were excluded from survival curve calculations, and their subsequent behavior was reported descriptively.

Analysis of the four failed cases revealed some shared clinical patterns. Three of the failures involved shorter (5 mm) abutments, and all occurred in posterior regions subject to higher occlusal load. Two failures were in the mandible and two in the maxilla, suggesting no strong arch dependency. All four cases were cemented with DentoTemp™ and presented broader occlusal tables or higher masticatory forces during follow-up. These common features may have compounded the intrinsic limitations of the provisional cement, contributing to premature loss of retention.

Kaplan–Meier survival analysis demonstrated cumulative survival rates of 95% for zirconia and 85% for metal–ceramic restorations at 36 months. Cement type strongly influenced survival: 100% for Fuji Plus^®^ versus 80% for DentoTemp™ (log-rank test not performed due to the very low number of events and the underpowered sample). In the Kaplan–Meier curves (Figure 6), loss of retention was considered the event at the time of failure; restorations re-cemented with Fuji Plus^®^ were not re-entered into the survival dataset, and their subsequent follow-up was reported descriptively. These differences should therefore be regarded as exploratory trends rather than inferential findings. Accordingly, the survival differences should be regarded exclusively as exploratory trends, not inferential findings.

### 3.4. Marginal Adaptation Evaluation

Marginal adaptation was initially assessed for all cases using conventional two-dimensional (2D) periapical and panoramic radiographs (Figure 7). Due to the intrinsic resolution limitations of these imaging modalities—approximately 150 μm for panoramic imaging and 75 μm for CBCT scans—precise quantitative measurements of marginal gaps could not be reliably performed using radiographs alone. No gross misfits, open margins, or overhanging contours were detected in any of the reviewed images, and all restorations were classified as radiographically acceptable, without apparent marginal discrepancies linked to early failure or peri-implant inflammation.

To supplement the radiographic evaluation, a subset of clinical images was analyzed using ImageJ software (Fiji version) to obtain semi-quantitative estimates of marginal gaps. Radiographic images were imported into ImageJ (NIH, Bethesda, MD, USA) for marginal-gap evaluation. Each image was calibrated to real dimensions using a visible reference, such as the known abutment diameter. The pixel-to-mm scale was set in ImageJ using this known reference dimension visible in each image. For each restoration, three points were selected along the visible crown–abutment interface within the 2D radiographic projection (typically near the mesial, central, and distal regions of the imaged side). Linear measurements were taken perpendicular to the interface at these locations to estimate the apparent gap width. Each time, three measurements were performed, and mean values were calculated after verifying consistency. The results were interpreted as qualitative indicators of marginal adaptation, suitable for comparative analysis between groups rather than absolute measurements. These ImageJ-derived values were interpreted strictly as semi-quantitative approximations, not true marginal-gap measurements, due to the intrinsic resolution limitations of 2D radiography.

Although these measurements were constrained by image resolution and calibration, they suggested a trend toward smaller marginal discrepancies in restorations cemented with Fuji Plus^®^ compared to those cemented with DentoTemp™. Representative radiographic examples from the study are presented in Figure 8.

As noted in the Methods, the OCT images included here are illustrative examples drawn from our archive and do not originate from the participants of this retrospective cohort.

In the existing literature, zirconia restorations [32,33] are generally associated with mean marginal fits of approximately 42 μm, while metal–ceramic restorations tend to exhibit slightly wider gaps of around 50 μm [34,35,36]. These values correspond with data from CAD/CAM prosthetic fabrication studies performed under controlled bonding and isolation protocols [37]. Although radiographic evaluation in the present study lacked the resolution to validate these differences, clinical outcomes were favorable, and no cases of cement-extrusion–related peri-implantitis were observed when standardized cementation protocols were followed.

An illustrative OCT example (Figure 9) is provided to demonstrate the high-resolution potential of this imaging modality for future investigations of marginal adaptation. OCT was not applied to the clinical sample in this study and is included here purely for reference. Despite the imaging limitations of 2D radiographs, no cases of cement-related peri-implant inflammation were observed when standardized cementation protocols were followed.

## 4. Discussion

This retrospective study evaluated the clinical performance of two luting agents, Fuji Plus^®^, a resin-modified glass ionomer, and DentoTemp™, a provisional acrylic–urethane-based cement, in partial implant-supported prostheses. The initial hypothesis, that Fuji Plus^®^ may provide higher retention than DentoTemp™, was supported by the descriptive trends observed in this cohort, but these findings must be interpreted cautiously given the retrospective design and limited statistical power.

### 4.1. Cement Performance

Fuji Plus^®^ demonstrated complete retention over the three-year follow-up, whereas DentoTemp™ presented four failures requiring re-cementation. Although this difference was clinically relevant, statistical analysis did not confirm significance (Fisher’s exact, *p* = 0.114), reflecting the limited subgroup sizes. The superior mechanical stability of Fuji Plus^®^ aligns with its classification as a definitive cement and supports its suitability for long-term use. In contrast, DentoTemp™, while originally designed as a provisional material, showed acceptable outcomes in selected cases, confirming its clinical utility when retrievability is desirable. However, this apparent difference should be interpreted strictly as an exploratory clinical trend rather than confirmatory evidence, given the retrospective, non-randomized design and limited statistical power.

During the three-year follow-up period, no major biological complications such as peri-implantitis or mucosal recession were observed. Mild inflammation occurred in four cases (10%), all of which coincided with radiographic evidence of subgingival cement remnants. These cases resolved following professional cleaning and localized debridement. This observation reinforces previous reports that residual cement is a significant risk factor for peri-implant inflammation [20,21,38,39,40,41], underscoring the importance of meticulous excess removal during cementation. Peri-implant soft tissue health is a decisive factor for long-term stability. Prior studies have shown that peri-implant lesions may be influenced not only by residual cement but also by loading strategies, with higher rates reported in immediately loaded implants compared with delayed protocols [42]. Although biological complications were limited, the chemical nature of the cement residue likely played a role in tissue response. While RMGI cements such as Fuji Plus^®^ and provisional acrylic–urethane cements differ in composition, the limited number of biological events observed in this study does not allow any comparative conclusions regarding biological behavior.

These findings are consistent with earlier studies that have emphasized the importance of cement selection in implant prosthodontics. However, our results should be interpreted as descriptive clinical trends rather than definitive comparative evidence. Larger, prospective trials comparing only definitive cements are necessary to strengthen clinical recommendations.

The decision to use DentoTemp™ in cases where retrievability was desirable introduces an element of clinical selection bias, since these cases may inherently present higher likelihood of early modification or esthetic refinement. As a result, the observed decementation frequency could reflect this practical indication rather than an intrinsic deficiency of the cement itself. While this mirrors real clinical reasoning, it may limit direct comparability between cement groups. Future prospective studies with pre-defined allocation criteria, for example, stratifying cases by indication for retrievability or abutment height, would help isolate the true material-related effect on retention. Clinically, these descriptive trends suggest that Fuji Plus^®^ performed more consistently in this cohort; however, definitive recommendations require larger prospective studies. DentoTemp™ remains suitable for specific scenarios, such as anterior restorations requiring possible shade adjustment, immediate delivery cases where occlusion is expected to evolve, or early loading phases in which retrievability is desirable for monitoring peri-implant tissues. In such contexts, the ease of debonding is advantageous and the risk of early recementation is clinically acceptable when patients are under close recall.

### 4.2. Comparative Analysis with Existing Literature

The survival and retention trends observed here are in line with prior clinical evidence for partial implant-supported restorations. For zirconia, Örtorp et al. reported 5-year survival of single crowns placed in private practice that is broadly comparable to our 36-month retention outcome (95%) [32]. Other clinical series have likewise shown high short- to mid-term survivals for zirconia restorations (e.g., 24–36 months) consistent with our findings [33]. With respect to marginal fit and adaptation, the literature commonly reports mean gaps on the order of ~42 µm for zirconia and ~50 µm for metal-ceramic frameworks under controlled fabrication and bonding protocols [34,35,36], which contextualizes the favorable biological performance observed here despite the limits of radiographic resolution.

Cement selection appears to be a stronger driver of retention than restorative material. Our data—complete retention with the resin-modified glass ionomer (Fuji Plus^®^) and all failures confined to the provisional acrylic-urethane cement (DentoTemp™)—mirror prior reports that resin-modified glass ionomers and definitive resin cements achieve higher and more consistent bond strength than provisional agents, with better long-term stability in implant prosthodontics [43,44]. Provisional cements remain useful where retrievability is prioritized, but the earlier debonding events seen in our cohort echo laboratory findings showing lower retentive strength for temporary materials under load [24,25,26,27,28,43,44]. From a mechanical reliability standpoint, these clinical patterns are coherent with broader fatigue-resistance data for implant components under cyclic loading [45].

Finally, our exploratory signal that taller abutments (7 mm) may enhance retention aligns with biomechanical principles and prior studies indicating that greater abutment height increases surface area and resistance to dislodgement [46,47]. While our subgroup is small, this interaction between abutment geometry and cement choice warrants confirmation in larger, prospective cohorts.

### 4.3. Abutment Height: 5 mm vs. 7 mm

Three of the four failures were associated with 5 mm abutments, compared to one with a 7 mm abutment. Although suggestive, the small number of failures limits the ability to draw statistically robust conclusions. The observed pattern aligns with biomechanical principles and in vitro evidence showing that increased abutment height enhances the surface area for adhesion and improves resistance to dislodgement [46,47]. These findings should therefore be regarded as exploratory. Future studies with larger cohorts are needed to confirm whether abutment height interacts with cement type to influence long-term retention outcomes. This is consistent with recent clinical data emphasizing how abutment geometry and customized transmucosal profiles can condition peri-implant soft-tissue stability and overall outcomes [48]. These preliminary findings justify a dedicated prospective investigation comparing abutment heights under standardized loading and cementation protocols to quantify the magnitude of this effect on long-term retention.

### 4.4. Limitations

It is important to recognize the various inherent limitations of this study. First, selection bias is introduced by the retrospective, non-randomized design, especially since DentoTemp™ was purposefully chosen in situations where retrievability was desired. Because of this, it is impossible to draw a causal conclusion because the cement groups were not equal at baseline. Second, there was insufficient statistical power to identify variations in retention or abutment-height effects due to the small sample size (*n* = 40), with only ten cases per subgroup. Third, because cement type was clearly recorded in the clinical records, evaluator blinding was not possible in a chart-based retrospective analysis. Nevertheless, the main outcome—loss of retention—is an objective clinical event with little room for subjective interpretation.

Additionally, care must be taken when interpreting the mechanical testing component. The laboratory experiment was specifically designed as an illustrative, exploratory supplement to the clinical findings rather than as a powered validation study, and repeated cementation cycles may introduce minor alterations to intaglio surfaces despite standardized cleaning.

The evaluation of the marginal gap was solely qualitative. The semi-quantitative ImageJ measurements were strictly interpreted as approximations rather than actual gap values, and radiographs lack the resolution to detect discrepancies below 75–150 μm. No quantitative or diagnostic conclusions can be made from OCT in this study because the images were only included as illustrative examples and were not taken from the clinical cases in this cohort.

Finally, all failures in the DentoTemp™ group were re-cemented with Fuji Plus^®^, which may have introduced a favorable bias for Fuji Plus^®^ during long-term follow-up. Collectively, these limitations justify interpreting the results as preliminary exploratory trends rather than definitive comparative evidence.

### 4.5. Future Directions

Future investigations should aim to validate these findings in larger, prospectively designed cohorts with longer follow-up periods. The systematic use of high-resolution imaging modalities like OCT provides more accurate and reproducible assessments of marginal adaptation beyond the limits of conventional radiography. In addition, analyses of abutment height and cement type may confirm their combined influence on prosthesis retention. Finally, further research on the biological response to different cements, especially regarding subgingival remnants, would help refine evidence-based guidelines for cement selection and clinical protocols in partial implant-supported restorations.

## 5. Conclusions

The current findings should be interpreted as preliminary observational trends rather than confirmatory evidence due to the inherent limitations of this small retrospective and non-randomized cohort. While all failures happened in cases that were initially cemented with the provisional material, Fuji Plus^®^ demonstrated a higher three-year retention than DentoTemp™. However, the two groups were not equal at baseline, and no causal conclusion can be made because cement selection was based on clinical judgment—particularly the use of DentoTemp™ in circumstances where retrievability was purposefully desired.

Biological issues were rare and mostly related to leftover cement, highlighting the significance of careful excess removal over material-specific effects. Although the mechanical and radiographic observations were in line with the clinical results, they should not be taken as official validation because they were exploratory in their purpose.

Overall, these results give proof to the potential benefits of Fuji Plus^®^ for long-term stability, while DentoTempTM might still be suitable in certain situations where retrievability is a top clinical concern. To validate these initial trends and more clearly define the indications for each luting agent in partial implant-supported restorations, larger prospective studies with standardized cement allocation and higher statistical power are needed.

## Figures and Tables

**Figure 1 medicina-61-02183-f001:**
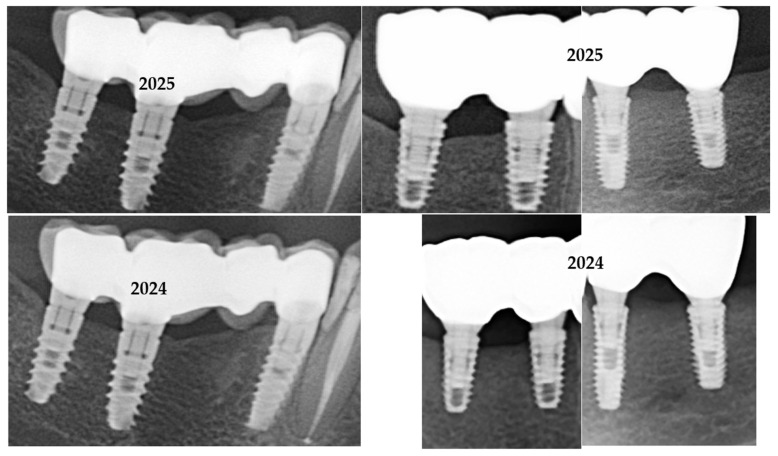
Sections of orthopantomographs (OPGs) showing two cemented partial restorations monitored between 2020 and 2025. The left panels depict a metal–ceramic three-unit prosthesis cemented with DentoTemp™, while the right panels show a zirconia three-unit prosthesis cemented with Fuji Plus^®^. Images are arranged chronologically from baseline (2020) to the most recent control (2025), illustrating stable peri-implant bone levels in both groups.

**Figure 2 medicina-61-02183-f002:**
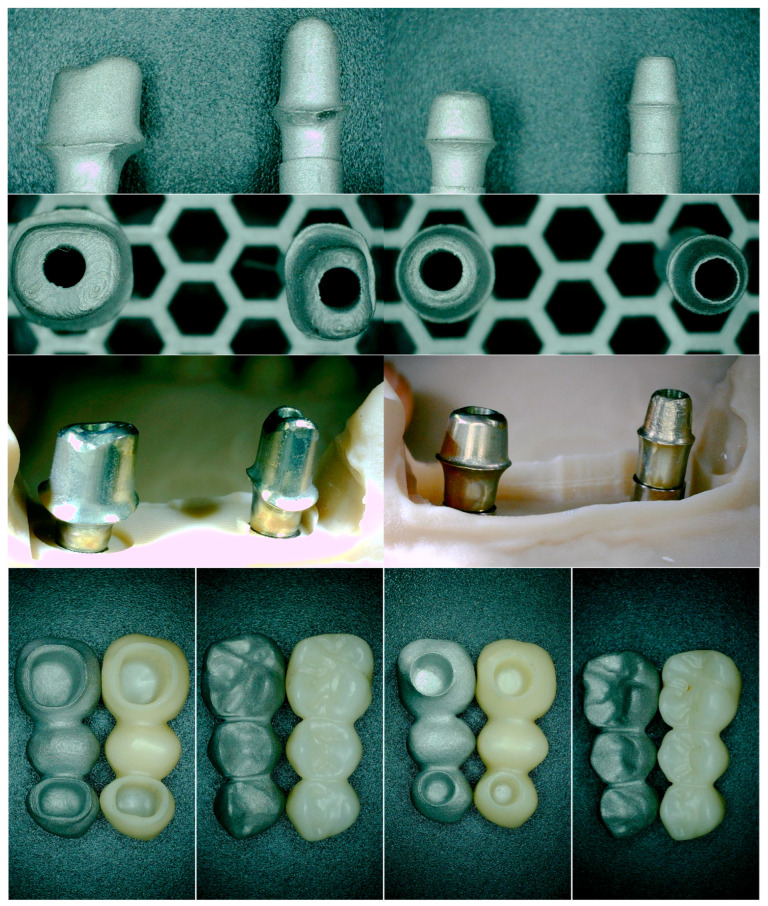
Representative abutments and partial restorations with two different abutment heights. The first row shows the lateral view of 7 mm (**left**) and 5 mm (**right**) abutments, while the second row presents the occlusal view of the same abutments. In the third row, the abutments are displayed within the working model from a lateral perspective. The fourth row illustrates zirconia and metal–ceramic three-unit restorations corresponding to the 7 mm (**left**) and 5 mm (**right**) abutments, viewed from both occlusal and basal aspects.

**Figure 3 medicina-61-02183-f003:**
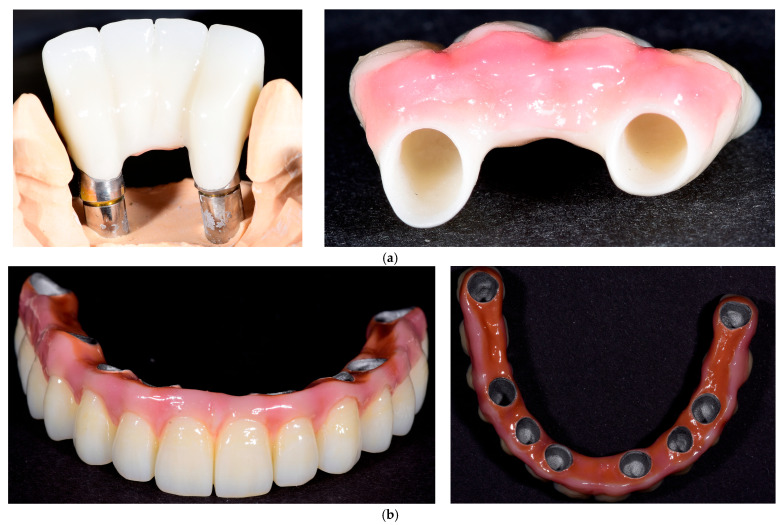
Examples of zirconia (**a**) and metal–ceramic (**b**) restorations. The zirconia samples illustrate partial fixed prostheses, while the metal–ceramic examples include a full-arch prosthesis shown only for material demonstration purposes. Full-arch cases were not part of the study cohort.

**Figure 4 medicina-61-02183-f004:**
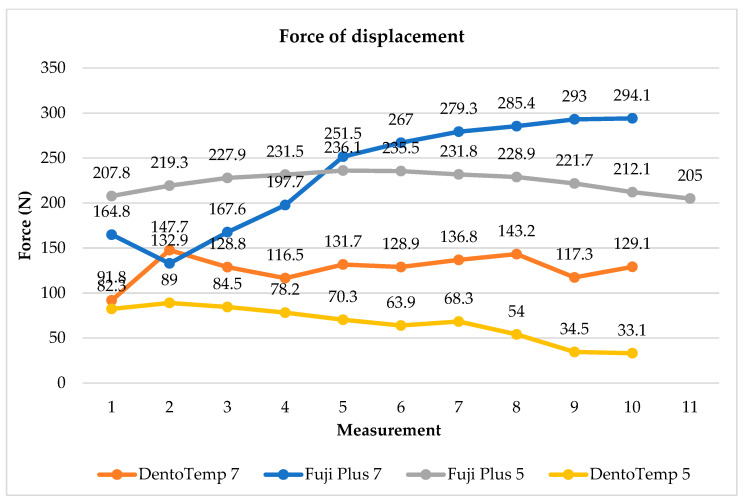
Force–displacement curves obtained during mechanical testing of cemented three-unit prostheses. Fuji Plus^®^ demonstrated higher peak retentive strength compared to DentoTemp™, and 7 mm abutments consistently outperformed 5 mm abutments.

**Figure 5 medicina-61-02183-f005:**
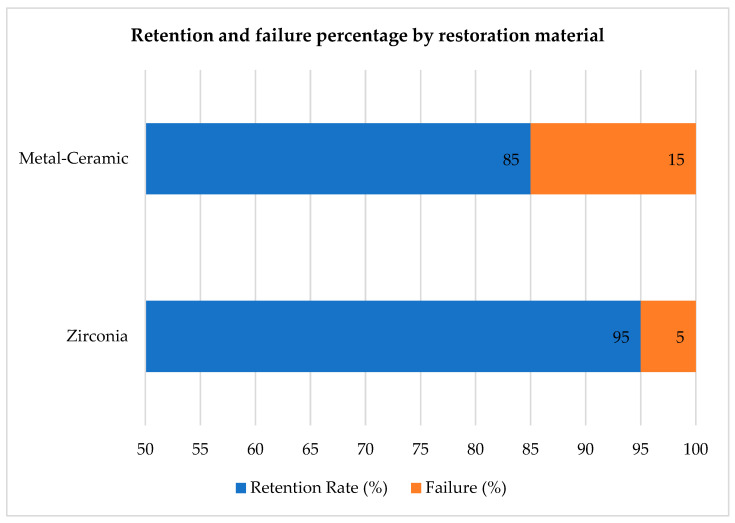
Stacked bar chart showing retention and failure percentages by restorative material. Zirconia restorations showed one failure (5%), while metal–ceramic restorations showed three failures (15%).

**Figure 6 medicina-61-02183-f006:**
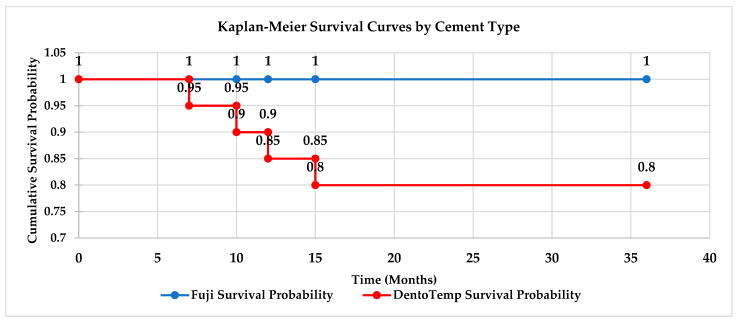
Kaplan–Meier survival curves for partial implant-supported restorations by cement type. Fuji Plus^®^ restorations maintained a 100% survival rate over 36 months, while DentoTemp™ restorations showed survival decreases at 7, 10, 12, and 15 months, stabilizing at 80% through the remainder of the follow-up.

**Figure 7 medicina-61-02183-f007:**
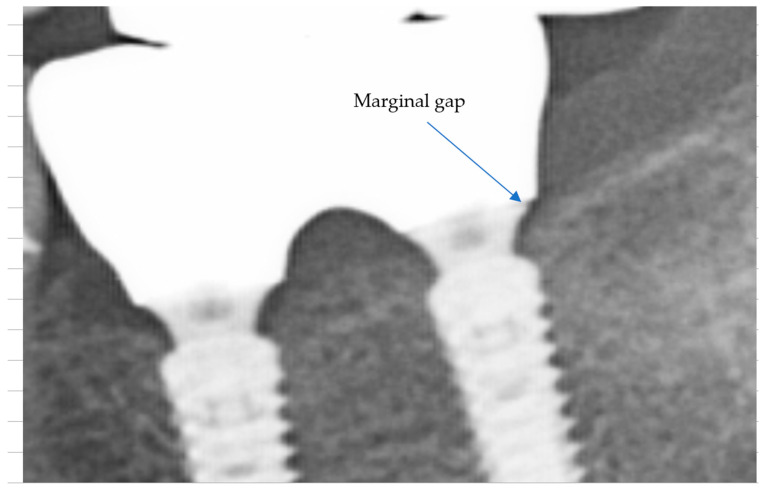
Section of an OPG showing a clearly visible large marginal gap. True marginal gaps in the range of 40–50 μm, relevant for clinical adaptation, cannot be visualized in conventional 2D imaging due to the inherent resolution limitations of panoramic radiographs (≥150 μm).

**Figure 8 medicina-61-02183-f008:**
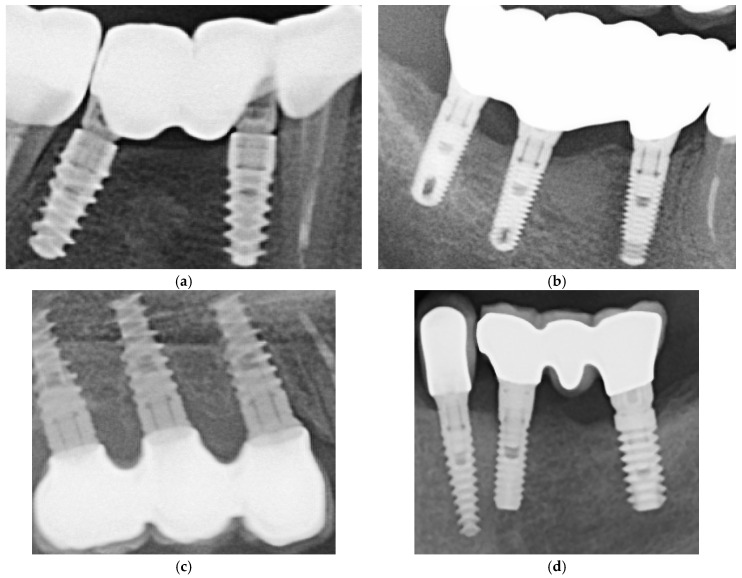
Sections of OPGs from study cases. The left panels (**a**,**c**,**e**,**g**) show restorations cemented with Fuji Plus^®^, while the right panels (**b**,**d**,**f**,**h**) show restorations cemented with DentoTemp™. Images illustrate representative follow-up cases used for radiographic evaluation of retention and marginal adaptation.

**Figure 9 medicina-61-02183-f009:**
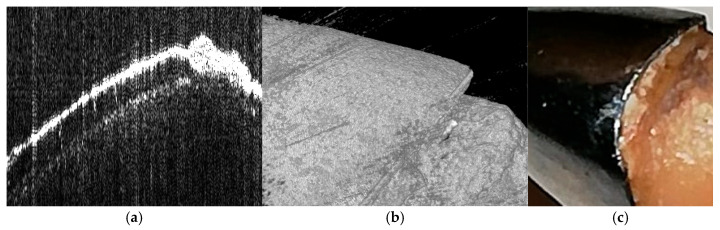
Illustrative examples of OCT imaging provided for reference. (**a**) OCT B-scan, (**b**) OCT 3D reconstruction, and (**c**) photograph of the investigated tooth. These images were included to demonstrate the potential of OCT for assessing marginal adaptation but were not obtained from the clinical cases analyzed in this study.

**Table 1 medicina-61-02183-t001:** Patient Demographics and Clinical Grouping According to Restoration Material and Cement Type.

Restoration Material	Cement Type	No. of Patients	Mean Age (Years)	Gender Distribution (M/F)	Maxillary Cases	Mandibular Cases
Zirconia	DentoTemp™	10	51.1	6/4	5	5
Zirconia	Fuji Plus	10	47.8	5/5	7	3
Metal-Ceramic	DentoTemp™	10	45.0	4/6	6	4
Metal-Ceramic	Fuji Plus	10	48.9	6/4	4	6

**Table 2 medicina-61-02183-t002:** Retention by Restoration Material.

Restoration Material	Total Cases	Failures	Retention Rate (%)
Zirconia	20	1	95
Metal-Ceramic	20	3	85

**Table 3 medicina-61-02183-t003:** Retention by Cement Type.

Cement Type	Total Cases	Failures	Retention Rate (%)
DentoTemp™	20	4	80
Fuji Plus	20	0	100

**Table 4 medicina-61-02183-t004:** Time-to-Failure per Patient.

Restoration Material	Abutment Height (mm)	Cement Type	Time to Failure (Months)	Re-Cemented with
Zirconia	5	DentoTemp™	12	Fuji Plus
Metal-Ceramic	5	DentoTemp™	7	Fuji Plus
Metal-Ceramic	5	DentoTemp™	10	Fuji Plus
Metal-Ceramic	7	DentoTemp™	15	Fuji Plus

## Data Availability

The original contributions presented in this study are included in the article. Further inquiries can be directed to the corresponding authors.

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
