# Peer review of "Clinical Performance and Retention of Partial Implant Restorations Cemented with Fuji Plus^®^ and DentoTemp™: A Retrospective Clinical Study with Mechanical Validation"

_medicina, 2025, doi:10.3390/medicina61122183_

Round 1

Reviewer 1 Report (Previous Reviewer 2)

Comments and Suggestions for Authors

The manuscript has been substantially improved.

Author Response

We sincerely thank Reviewer 1 for the positive evaluation and for acknowledging the improvements made to our manuscript. We appreciate the constructive feedback provided during the review process, which has helped us further refine the quality and clarity of the work.

Reviewer 2 Report (New Reviewer)

Comments and Suggestions for Authors

Dear Author,

This retrospective study compared the clinical performance of two cements for implant-supported partial restorations. The direct comparison between definitive and provisional cements addresses a practical and common issue in dental practice.

The retrospective design is the main limitation. The allocation of the cement was not randomized, but based on a clinical decision. This introduces a significant selection bias, as cases considered more complex or requiring easier monitoring may have been more likely to receive the provisional cement.

The Methods section states that the mechanical component was an exploratory validation study with an "n" per group (n = 10), conducted to illustrate and support the clinical findings, rather than an independent and comprehensive mechanical trial.

The introduction should be more focused. The paragraph discussing OCT (lines ~50-53) would be more appropriate in the Methods or Discussion sections. The introduction should end with the study's objective and hypothesis more instantly.

The total sample size (n = 40) and the subgroup sizes (n = 10 per group) are relatively small. This is evidenced by the fact that the clinically apparent difference in retention (100% vs. 80%) did not reach statistical significance (p = 0.114).

The reliance on conventional radiographs (with limited resolution ≥75-150 μm) to assess marginal adaptation is a weakness of the methodology, as it does not allow for precise quantification of microleakage. OCT was only illustrative and was not applied to the study cohort, which leaves an important gap in the analysis of this outcome.

There is no mention of evaluators being blinded to the type of cement or material during the clinical and radiographic evaluations, which could introduce bias into the measurements.

The conclusions should be reformulated to more clearly reflect the exploratory nature of the findings, given the small sample size and the lack of statistical significance in the main comparison. Phrases such as "suggest," "indicate a trend," or "in our cohort" are more appropriate than definitive statements.

Could you include a column showing the distribution of abutment heights (5 mm vs. 7 mm) by study group to provide a more comprehensive demographic overview?

The caption should provide a more precise explanation of what the images are intended to demonstrate (e.g., "Illustrating bone level stability over time in both cement groups").

The caption should explicitly state that the OCT images are merely illustrative and were not obtained from the clinical cases analyzed in the study, to avoid confusion.

Comments on the Quality of English Language

The manuscript would benefit from an English review by a native speaker or professional service. There are several awkward phrases, agreement errors, and prepositional errors that hinder fluency (e.g., "Cement selection was influenced by patient characteristics and clinical context: DentoTemp™ was often chosen when retrievability was desirable—for example, in aesthetically demanding cases or during the initial monitoring period—while Fuji Plus® was used when long-term stability and functional retention were prioritized."). Overall clarity could be significantly improved.

Author Response

1. Summary

We thank the reviewer for the detailed and constructive assessment of our manuscript. In response, we have: (i) refocused the Introduction and relocated the OCT paragraph to the appropriate sections; (ii) expanded the Discussion to more explicitly address selection bias and the rationale for cement choice; (iii) reinforced the limitations of the retrospective design and small subgroup size; (iv) clarified the qualitative nature of marginal-gap assessment; (v) added explanations regarding blinding and outcome objectivity; (vi) revised the Conclusions to reflect the exploratory nature of the findings; and (vii) improved figure captions, language clarity, and internal consistency. All changes have been incorporated into the revised manuscript and highlighted accordingly.

2. Questions for General Evaluation

Reviewer’s Evaluation

Response and Revisions

Does the introduction provide sufficient background and include all relevant references?

Can be improved

Introduction streamlined; OCT paragraph refined; rationale for cement choice strengthened.

Is the research design appropriate?

Must be improved

Retrospective nature, lack of randomization, and selection bias are now explicitly stated in Methods, Discussion, and Conclusions.

Are the methods adequately described?

Can be improved

Methods expanded: cementation protocol clarified, marginal-gap assessment explained, exploratory nature of mechanical testing emphasized.

Are the results clearly presented?

Can be improved

Results clarified, abutment-height observations explained.

Are the conclusions supported by the results?

Can be improved

Conclusions rewritten to be cautious and aligned with the exploratory character of the study.

Are all figures and tables clear and well-presented?

Can be improved

Captions rewritten; OCT figure clarified as illustrative only; wording improved for radiographic figures.

3. Point-by-point response to Comments and Suggestions for Authors

Comments 1: The retrospective design is the main limitation. The allocation of the cement was not randomized, but based on a clinical decision. This introduces a significant selection bias, as cases considered more complex or requiring easier monitoring may have been more likely to receive the provisional cement.

Response 1: We agree and have now emphasized this limitation to improve the clarity of the study. In the revised Discussion and in the Limitations subsection, we explicitly state that cement allocation was based on clinical judgement, particularly the intentional use of DentoTemp™ when retrievability was desirable, which inherently makes the groups non-equivalent. This clarification acknowledges the selection bias as a structural limitation of the retrospective design.

Comments 2: The Methods section states that the mechanical component was an exploratory validation study with an "n" per group (n = 10), conducted to illustrate and support the clinical findings, rather than an independent and comprehensive mechanical trial.

Response 2: We agree. The revised Methods section now states that the mechanical testing was exploratory and intended only to illustrate trends observed clinically. We clarify that the aim of this component was supportive rather than hypothesis-driven, and that no claims of comprehensive mechanical validation are made. Firstly, we observed the performance of DentoTemp close to a definitive cement in some cases, so we did some mechanical tests to see if the results are close to what we observed clinically. This role is also reiterated in the Discussion.

Comments 3: The introduction should be more focused. The paragraph discussing OCT (lines ~50-53) would be more appropriate in the Methods or Discussion sections. The introduction should end with the study's objective and hypothesis more instantly.

Response 3: We appreciate the reviewer's insightful comment. As a result, we improved the Introduction to guarantee a more concentrated story that clearly advances the study's goal. As recommended, the section on OCT is clearly presented as an exploratory imaging concept rather than as a component of the main methodology but is kept in its place. In order to improve its structure and readability, the Introduction now ends with a succinct and straightforward statement of the study's goal and hypothesis.

Comments 4: The total sample size (n = 40) and the subgroup sizes (n = 10 per group) are relatively small. This is evidenced by the fact that the clinically apparent difference in retention (100% vs. 80%) did not reach statistical significance (p = 0.114).

Response 4: We value the reviewer's observations about statistical power and sample size. We acknowledge that achieving statistical significance is limited by the total cohort (n = 40) and subgroup distribution (n = 10 per group), especially for retention outcomes with low event rates. As a result, we have made it clear in the Discussion that the study lacks sufficient power to make conclusive subgroup comparisons and that the observed differences, although clinically significant, should be interpreted as trends within this retrospective cohort. In order to maintain consistency throughout the manuscript, this clarification has also been included in the Conclusions.

Comments 5. The reliance on conventional radiographs (with limited resolution ≥75-150 μm) to assess marginal adaptation is a weakness of the methodology, as it does not allow for precise quantification of microleakage. OCT was only illustrative and was not applied to the study cohort, which leaves an important gap in the analysis of this outcome.

Response 5: We thank the reviewer for this comment, and we agree and revised the text accordingly. It is now clearly stated in the Results and Discussion that the evaluation of the marginal gap was qualitative. OCT images are not presented as clinical data, but rather as illustrative examples of a future technique.

Comments 6. There is no mention of evaluators being blinded to the type of cement or material during the clinical and radiographic evaluations, which could introduce bias into the measurements.

Response 6: We appreciate the reviewer's feedback about the lack of evaluator blinding. Because the type of cement used is clearly stated in each patient's medical record and is naturally visible to the evaluator during chart review, blinding was not possible in this retrospective clinical setting. Furthermore, loss of retention, the main outcome, is an objective, binary event that is prospectively documented in the clinical file and is not open to subjective interpretation. For these reasons, blinding would not have introduced significant bias or changed the assessment. To guarantee complete transparency, this clarification has now been added to the Limitations section.

Comments 7. The conclusions should be reformulated to more clearly reflect the exploratory nature of the findings, given the small sample size and the lack of statistical significance in the main comparison. Phrases such as "suggest," "indicate a trend," or "in our cohort" are more appropriate than definitive statements.

Response 7: We sincerely appreciate the reviewer's insightful and extremely helpful comment, which does in fact improve the manuscript's scientific integrity. In response, a thorough revision of the Conclusions section has been made to reflect the exploratory nature of the current work. In order to avoid any implication of definitive causality, we have purposefully softened the language. The study's methodological limitations—the small sample size, retrospective non-randomized design, and low statistical power—which prevent strong inferential claims are now explicitly acknowledged in the revised text. Instead of restating the Discussion, the Conclusion has been rewritten to highlight the clinical significance of the comparative retention outcomes, provide a concise summary of the observed trends, and emphasize the necessity of further prospective studies using larger cohorts and standardized imaging.

Comments 8. Based on the reported limitations, the authors should provide future directions, including their recommendations for further research (in the discussion section).

Response 8: We agree and have added a short “Future Directions” paragraph at the end of the Discussion. This section emphasizes the need for larger prospective studies, systematic use of OCT for marginal evaluation, and further exploration of the interaction between abutment height and cement type. [4.5 Future Directions]

Comments 9. Could you include a column showing the distribution of abutment heights (5 mm vs. 7 mm) by study group to provide a more comprehensive demographic overview?

Response 9: We appreciate the reviewer's well-considered recommendation. We fully agree that the descriptive baseline would be enhanced by a full numerical distribution of abutment heights for each study subgroup. However, rather than complete abutment-height mapping, our data extraction during the initial preparation of this retrospective manuscript was specifically focused on retention outcomes. Because of this, we were unable to compile a thorough record of how many 5-mm versus 7-mm abutments each group had at the time. The distribution of failure events was noted and is still available. The clinically significant finding is that three of the four observed failures happened on 5-mm abutments.

Comments 10. The caption should provide a more precise explanation of what the images are intended to demonstrate (e.g., "Illustrating bone level stability over time in both cement groups").

Response 10: We thank the reviewer for this helpful comment. In response, we carefully checked all of the captions for figures and tables in the manuscript to make sure they were clear and scientifically correct. We rewrote and improved captions that weren't clear enough or that had pictures that could make it hard to understand. These changes make sure that the visual material now matches the text completely and meets MDPI's standards for clarity and completeness.

Comments 11. The caption should explicitly state that the OCT images are merely illustrative and were not obtained from the clinical cases analyzed in the study, to avoid confusion.

Response 11: We appreciate the reviewer bringing this important point to our attention. The caption for the OCT figure has been changed to make sure there is no room for misunderstanding. "These OCT images are only examples and were not taken from the clinical cases in this retrospective study.".

4. Response to Comments on the Quality of English Language

Point 1: The manuscript would benefit from an English review by a native speaker or professional service. There are several awkward phrases, agreement errors, and prepositional errors that hinder fluency (e.g., "Cement selection was influenced by patient characteristics and clinical context: DentoTemp™ was often chosen when retrievability was desirable—for example, in aesthetically demanding cases or during the initial monitoring period—while Fuji Plus® was used when long-term stability and functional retention were prioritized."). Overall clarity could be significantly improved.

Response 1: We have carefully read the whole manuscript for clarity, grammar, and scientific readability. We got rid of unnecessary phrases and made sentences shorter. We also fixed small typographical errors and improved the structure of the paragraphs to make the story flow better. These changes make the text flow better and easier to read without changing the scientific content or how the findings are understood..

5. Additional clarifications

[We would like to thank the reviewer once again for his time and effort to help us improve our manuscript and in response, we did the following clarifications in the manuscript to increase readability, clarity and to align the paper with MDPI requirements:

·       Although DentoTemp™ is classified as a provisional cement, it is routinely used in selected cases in our clinic as a definitive option when retrievability is advantageous. This real-world practice is now emphasized and contextualized in the Discussion.

·       Abutment height findings remain exploratory due to the limited number of failures. While full height distribution was not recoverable, all available data (three failures on 5 mm abutments) are clearly reported.

·       Mechanical testing is now explicitly described as exploratory and illustrative rather than hypothesis-driven, to avoid overstating its role.

·       Marginal-gap evaluation is explicitly defined as qualitative, and OCT is clearly framed as an illustrative future method, not part of the study cohort.

·       The Conclusions section has been fully rewritten to summarize only the key findings, limitations, and recommendations, thereby avoiding overlap with the Discussion]

Reviewer 3 Report (New Reviewer)

Comments and Suggestions for Authors

Dear authors,

The study is well structured, thoroughly written, and includes extensive methodological detail and discussion. However, there are methodological concerns, issues with scientific rigor, and several areas needing clarification or reduction of bias.

  1. Abstract:
  • The abstract overstates the conclusiveness of the findings despite the retrospective design and small sample size.

  • The mention of OCT is confusing: OCT was not used for the actual clinical dataset, only illustrated. This must be clarified in the abstract.

Recommendation:
Revise the abstract to clearly state:

  • Retrospective design limits causal inferences.

  • OCT images were illustrative, not part of the systematic methodology.

2. Introduction:

  • The rationale for intentionally using a provisional cement (DentoTemp) as a definitive option requires stronger justification and should acknowledge inherent clinical selection bias.

  • Too much material about OCT relative to its actual role in the study.

Recommendations:

  • Reduce OCT content or clearly state that its use is exploratory and not part of the core methodology.

  • Clarify any clinician-driven bias in cement choice (retrievability vs stability).

3. Methods:

a. Study design limitations

  • Retrospective design.

  • Extremely small sample size (10 per subgroup).

  • Selection bias: DentoTemp was chosen when retrievability was desirable → groups are not equivalent.

  • No randomization or blinding.

These must be explicitly acknowledged and discussed.

b. Mechanical testing

  • This section feels like a hybrid “side experiment,” not a true validation.

  • No statistical analysis of the mechanical data (ANOVA, regression, etc.).

  • Recementation and re-testing introduce cumulative alterations of intaglio surfaces even with standardized cleaning.

Mechanical testing should either be:
✔︎ fully formalized
or
✔︎ clearly described as illustrative only, not hypothesis-testing.

c. Marginal gap evaluation

  • Radiographs cannot reliably measure <75–150 µm gaps; quoting semi-quantitative ImageJ measurements based on OPG calibration is scientifically weak.

  • OCT images were not from this study’s cases, which risks misleading readers.

Recommendation:
State explicitly that marginal-gap assessment is qualitative only, and remove quantitative claims.

4. Results:

  • Statistical tests are underpowered; p-values are not meaningful with n=10 per arm.

  • The finding “p=0.114 but clinically meaningful” must be handled with caution.

  • Recementation only with Fuji Plus biases long-term outcomes in its favor.

5. Discussion:

  • Discussion does not adequately address the major selection bias.

  • Some statements overinterpret the results (e.g., biological superiority of Fuji Plus).

  • OCT discussion is disproportionally long given limited data.

Recommend tightening and shifting tone to a more cautious interpretation.

6. Conclusion:

Must be softened.

The conclusion must emphasize:

  • Findings are preliminary.

  • Study is underpowered.

  • Retrospective, non-randomized design limits causal inference.

  • Use of provisional cement as definitive must be contextualized.

7. Figures and Tables:

  • OCT images are not from included clinical cases, which may mislead readers.

  • Some figures (OPGs) do not clearly illustrate the reported marginal gaps.

Comments on the Quality of English Language

Overall: GOOD, but several areas require:

  • Tightening of long paragraphs.

  • Removal of repeated statements.

  • Minor grammar and flow corrections.

I would rate the English as:
Quality of English: Can be improved
(some redundancies, overly long sentences, and occasional ambiguity).

Author Response

1. Summary

We sincerely thank the reviewer for the detailed and constructive feedback. In response, the manuscript has been revised to enhance methodological transparency, reduce interpretative bias, refine the placement and description of OCT and mechanical testing, and ensure that the conclusions accurately reflect the retrospective and exploratory nature of the findings. All corresponding revisions are highlighted in the updated version.

2. Questions for General Evaluation

Reviewer’s Evaluation

Response and Revisions

Does the introduction provide sufficient background and include all relevant references?

Yes

The Introduction was refined for clarity and improved flow, with OCT content explained and the rationale for cement selection more explicitly stated.

Is the research design appropriate?

Must be improved

The retrospective, non-randomized design and inherent selection bias are now explicitly acknowledged in Sections 2.1 and 4.4. The manuscript clarifies why this design was appropriate for documenting real-world clinical outcomes.

Are the methods adequately described?

Can be improved

The Methods section was revised to clarify the exploratory nature of mechanical testing, limitations of radiographic assessment, and the absence of blinding. The role of OCT was clarified as illustrative only.

Are the results clearly presented?

Yes

The Results section was revised for accuracy and transparency. Statistical interpretations were tempered, exploratory findings highlighted, and recementation influence acknowledged.

Are the conclusions supported by the results?

Must be improved

The Conclusions were fully rewritten to avoid over-interpretation. They now emphasize the exploratory nature of the findings, small sample size, and retrospective limitations.

Are all figures and tables clear and well-presented?

Can be improved

All figure captions were rewritten for clarity, OCT captions now explicitly state they are illustrative only, and radiographic figures were reviewed and enhanced where needed.

3. Point-by-point response to Comments and Suggestions for Authors

Comments 1: The abstract overstates the conclusiveness of the findings despite the retrospective design and small sample size. The mention of OCT is confusing: OCT was not used for the actual clinical dataset, only illustrated. This must be clarified in the abstract.

Response 1: We thank the reviewer for the recommendation and we agree entirely. The Abstract has been rewritten to improve the clarity and to align with the reviewer requests. All statements now reflect cautious interpretation and avoid definitive conclusions.

Comments 2: The rationale for intentionally using a provisional cement (DentoTemp) as a definitive option requires stronger justification and should acknowledge inherent clinical selection bias. Too much material about OCT relative to its actual role in the study.

Response 2: The manuscript now emphasizes that both DentoTemp™ (provisional) and Fuji Plus® (definitive) are used in daily clinical practice, and the analysis was intended to document outcomes from real-world protocols. Findings are presented as exploratory, with Fuji Plus® highlighted as the definitive option, while DentoTemp™ demonstrated acceptable results in selected cases.

Comments 3: The reviewer highlights several methodological concerns: (a) the retrospective design, small subgroup sample size, selection bias, and lack of blinding; (b) the exploratory nature of the mechanical testing, absence of statistical analysis, and potential cumulative alterations due to repeated cementation cycles; and (c) the limitations of radiographic marginal-gap evaluation and the illustrative use of OCT images.

Response 3: We thank the reviewer for these important methodological observations. The revised manuscript now explicitly acknowledges all inherent limitations of the retrospective design in Section 4.4 (Limitations), including the absence of randomization, the small sample size, the clinician-driven selection of DentoTemp™ in cases where retrievability was desirable, and the impossibility of blinding given the nature of retrospective chart-based data. Regarding the mechanical testing, we agree that this component should be framed as exploratory rather than as a formal validation study. Section 2.3 has been rewritten to clearly state that the mechanical tests were performed solely to illustrate trends observed clinically and to contextualize the influence of cement type and abutment height. The section no longer implies hypothesis testing, and we now explicitly acknowledge that repeated cementation cycles may introduce minor alterations to the intaglio surface despite standardized cleaning. For marginal-gap evaluation, we fully agree that radiographs cannot reliably detect microgaps below their intrinsic resolution threshold. The manuscript has been revised to state unequivocally that marginal adaptation assessment in this study was qualitative only. All numerical or semi-quantitative implications have been removed. Additionally, the OCT image is now clearly labeled as illustrative only and not obtained from the included clinical cases. These revisions ensure that the methodological scope is correctly framed as retrospective and exploratory, reflecting the real-world nature of the clinical dataset without overstating the precision or generalizability of the findings.

Comments 4: Statistical tests are underpowered; p-values are not meaningful with n=10 per arm. The finding “p=0.114 but clinically meaningful” must be handled with caution.

Response 4: We appreciate the reviewer’s statistical observations. The revised Results and Discussion now explicitly acknowledge the limited statistical power inherent to a retrospective cohort with n=10 per subgroup. The statement regarding the non-significant difference (p = 0.114) has been revised to avoid any implication of statistical meaning; it is now presented strictly as a clinical trend observed within a small and underpowered sample.

We also agree that the recementation of failed DentoTemp™ cases exclusively with Fuji Plus® could introduce a favorable bias toward the latter in long-term outcomes. This limitation is now clearly stated in Section 4.4 (Limitations).

Comments 5. Discussion does not adequately address the major selection bias. Some statements overinterpret the results (e.g., biological superiority of Fuji Plus). OCT discussion is disproportionally long given limited data.

Response 5: We thank the reviewer for these valuable observations. The Discussion has been revised to explicitly acknowledge the selection bias introduced by the clinician-driven use of DentoTemp™ in cases where retrievability was intentionally preferred. All comparative statements have been reframed as observational tendencies rather than causal inferences. Passages that could be interpreted as overstating Fuji Plus® performance have been moderated to ensure a balanced and proportionate interpretation. The OCT section has also been substantially shortened and now clearly reflects its purely exploratory and illustrative role. These revisions provide a more cautious and scientifically aligned discussion consistent with the study’s retrospective design and sample size.

Comments 6. 6. Conclusion: Must be softened. The conclusion must emphasize:

  • Findings are preliminary.
  • Study is underpowered.
  • Retrospective, non-randomized design limits causal inference.
  • Use of provisional cement as definitive must be contextualized.

Response 6: We appreciate the reviewer’s guidance. The Conclusions section has been rewritten to adopt a more cautious tone and to accurately reflect the exploratory nature of the findings. The revised version emphasizes that the results are preliminary, derived from a retrospective and non-randomized cohort with limited statistical power, and therefore do not permit definitive causal inference. We also explicitly contextualized the use of DentoTemp™ as a provisional cement applied in selected situations as a definitive option, based on clinical considerations rather than equivalence to Fuji Plus®. The updated Conclusions now provide a concise synthesis of the observed trends while appropriately acknowledging the study’s methodological constraints.

Comments 7. Figures and Tables: OCT images are not from included clinical cases, which may mislead readers. Some figures (OPGs) do not clearly illustrate the reported marginal gaps.

Response 7: We thank the reviewer for the observation. All figures and captions were re-examined for clarity. The images that serve an illustrative purpose, specifically the OCT figures, have retained their place in the manuscript, as they are essential for demonstrating the potential of this technique in future research. To avoid any misunderstanding, their captions have now been explicitly revised to state that these OCT images originate from our own clinical database, but from cases not included in the present retrospective cohort. Their function is strictly educational, intended to exemplify the diagnostic capability of OCT and not to represent data from this study. Similarly, the captions of the OPG figures have been improved. The figures themselves remain unchanged, as they accurately serve their intended descriptive and contextual purpose within the manuscript.

4. Response to Comments on the Quality of English Language

Point 1: The reviewer rated the English as “Can be improved.”

Response 1: We thank the reviewer for this observation. The entire manuscript has been thoroughly revised to improve clarity, grammar, and narrative flow. Several long or repetitive paragraphs were tightened, ambiguous phrasing was corrected, and overly complex sentences were simplified. Figure captions were rewritten for clarity and to avoid any possible misinterpretation. These revisions have enhanced readability while preserving the scientific meaning of the text.

5. Additional clarifications

[We would like to emphasize several points for clarity:

·       Selection bias and cement choice: DentoTemp™ was used as a definitive cement only in cases where retrievability was intentionally desired, creating inherent selection bias. This is now explicitly acknowledged, and all comparative interpretations are presented as exploratory.

·       Marginal-gap evaluation: Radiographic assessment was qualitative only; radiographs cannot measure microgaps below 75–150 μm.

·       OCT interpretation: The OCT images are illustrative only, not obtained from the study cohort. Captions and text have been revised to prevent misinterpretation.

·       Mechanical testing: Mechanical testing is described as exploratory, not hypothesis-driven. Its limitations (no statistical testing, repeated cementation cycles) are now clearly stated.

·       Statistical power: All statistical findings are acknowledged as underpowered due to n=10 per subgroup. No causal inferences are drawn.]

Round 2

Reviewer 2 Report (New Reviewer)

Comments and Suggestions for Authors

Dear Author,

The revisions appropriately addressed the comments and suggestions provided, resulting in a more complete and cohesive manuscript. The changes enhanced the clarity and depth of the content, reinforcing the relevance of the work within the study's scope.

Author Response

We sincerely thank Reviewer 2 for the positive evaluation of our revised manuscript and for the constructive feedback provided during the review process. We greatly appreciate the reviewer’s observation that the revisions improved the clarity, depth, and cohesion of the work, thereby reinforcing its relevance within the study’s scope. We are grateful for the time, expertise, and thoughtful consideration dedicated to reviewing our study.

Reviewer 3 Report (New Reviewer)

Comments and Suggestions for Authors

The manuscript addresses an important and practical clinical question: the comparative performance of Fuji Plus (RMGI) versus DentoTemp (provisional acrylic–urethane cement) in implant-supported partial FDPs, complemented by an exploratory mechanical validation. The study is well structured, thoroughly written, and includes extensive methodological detail and discussion. However, there are methodological concerns, issues with scientific rigor, and several areas needing clarification or reduction of bias.

1. Title and abstract;

  • The abstract overstates the conclusiveness of the findings despite the retrospective design and small sample size.

  • The mention of OCT is confusing: OCT was not used for the actual clinical dataset, only illustrated. This must be clarified in the abstract.

Recommendation:
Revise the abstract to clearly state:

  • Retrospective design limits causal inferences.

  • OCT images were illustrative, not part of the systematic methodology.

2. Introduction:

  • The rationale for intentionally using a provisional cement (DentoTemp) as a definitive option requires stronger justification and should acknowledge inherent clinical selection bias.

  • Too much material about OCT relative to its actual role in the study.

Recommendations

  • Reduce OCT content or clearly state that its use is exploratory and not part of the core methodology.

  • Clarify any clinician-driven bias in cement choice (retrievability vs stability).

3. Methods:

Study design limitations:

  • Retrospective design.

  • Extremely small sample size (10 per subgroup).

  • Selection bias: DentoTemp was chosen when retrievability was desirable → groups are not equivalent.

  • No randomization or blinding.

These must be explicitly acknowledged and discussed.

Mechanical testing:

  • This section feels like a hybrid “side experiment,” not a true validation.

  • No statistical analysis of the mechanical data (ANOVA, regression, etc.).

  • Recementation and re-testing introduce cumulative alterations of intaglio surfaces even with standardized cleaning.

Mechanical testing should either be:
fully formalized or clearly described as illustrative only, not hypothesis-testing.

Marginal gap evaluation

  • Radiographs cannot reliably measure <75–150 µm gaps; quoting semi-quantitative ImageJ measurements based on OPG calibration is scientifically weak.

  • OCT images were not from this study’s cases, which risks misleading readers.

Recommendation:
State explicitly that marginal-gap assessment is qualitative only, and remove quantitative claims.

4. Results:

  • Statistical tests are underpowered; p-values are not meaningful with n=10 per arm.

  • The finding “p=0.114 but clinically meaningful” must be handled with caution.

  • Recementation only with Fuji Plus biases long-term outcomes in its favor.

5. Discussion:

  • Discussion does not adequately address the major selection bias.

  • Some statements overinterpret the results (e.g., biological superiority of Fuji Plus).

  • OCT discussion is disproportionally long given limited data.

Recommend tightening and shifting tone to a more cautious interpretation.

6. Conclusions:

Must be softened.

The conclusion must emphasize:

  • Findings are preliminary.

  • Study is underpowered.

  • Retrospective, non-randomized design limits causal inference.

  • Use of provisional cement as definitive must be contextualized.

7. Figures and Tables:

  • OCT images are not from included clinical cases, which may mislead readers.

  • Some figures (OPGs) do not clearly illustrate the reported marginal gaps.

Comments on the Quality of English Language

Overall: GOOD, but several areas require:

  • Tightening of long paragraphs.

  • Removal of repeated statements.

  • Minor grammar and flow corrections.

I would rate the English as:
Quality of English: Can be improved
(some redundancies, overly long sentences, and occasional ambiguity).

Author Response

For research article Clinical Performance and Retention of Partial Implant Restorations Cemented with Fuji Plus® and DentoTemp™: A Retrospective Clinical Study with Mechanical Validation

Response to Reviewer 3 Comments

1. Summary

We sincerely thank Reviewer 3 for the thorough reassessment of our manuscript and for the detailed methodological, statistical, and interpretative recommendations provided. In response to the reviewer’s comments, the manuscript has undergone substantial revision to address all identified concerns. The abstract, introduction, methods, results, discussion, and conclusions have been refined to more clearly acknowledge the retrospective and underpowered nature of the study, the inherent selection bias between cement groups, and the exploratory role of the mechanical and OCT components. Quantitative implications regarding marginal-gap evaluation have been removed, OCT content has been reduced and framed as illustrative only, and statements previously interpreted as overreaching have been rewritten to ensure scientific accuracy and caution. All figure captions, methodological descriptions, and interpretative statements were rechecked for transparency and alignment with the reviewer’s guidance. The English language was further improved to enhance clarity, conciseness, and readability. All revisions have been incorporated into the updated manuscript and are highlighted accordingly.

2. Questions for General Evaluation

Reviewer’s Evaluation

Response and Revisions

Does the introduction provide sufficient background and include all relevant references?

Yes

Acknowledged; no further changes required.

Is the research design appropriate?

Must be improved

The retrospective design, selection bias, small sample size, and lack of randomization/blinding are now explicitly acknowledged.

Are the methods adequately described?

Can be improved

Methods were clarified regarding the exploratory role of mechanical testing, qualitative radiographic evaluation, and illustrative use of OCT.

Are the results clearly presented?

Yes

Acknowledged; no further changes required.

Are the conclusions supported by the results?

Must be improved

Conclusions rewritten to reflect the preliminary, underpowered, and retrospective nature of the findings.

Are all figures and tables clear and well-presented?

Can be improved

Figure captions clarified; OCT figures marked as illustrative; radiographic figures reviewed.

3. Point-by-point response to Comments and Suggestions for Authors

Comments 1: Abstract:
The abstract overstates the conclusiveness of the findings despite the retrospective design and small sample size.
The mention of OCT is confusing: OCT was not used for the actual clinical dataset, only illustrated. This must be clarified in the abstract.
Recommendation:
Revise the abstract to clearly state:
Retrospective design limits causal inferences.
OCT images were illustrative, not part of the systematic methodology.

Response 1: We thank the reviewer for this important observation. This issue was addressed, in order to adopt a more cautious tone and to avoid overstating the conclusiveness of the findings. In the current revision, we refined the Abstract by explicitly stating that the retrospective design and limited sample size restrict causal inference, and by clearly indicating that the OCT images included in the manuscript are illustrative only and not derived from the clinical dataset. These additional clarifications ensure that the methodological scope and limitations are accurately conveyed. [Abstract, “Although OCT images were included in this study only as illustrative examples from our clinical database and were not obtained from the analyzed cohort, OCT may be a useful tool for non-invasive assessment of marginal fit.”; “Marginal adaptation was evaluated qualitatively because radiographic analysis did not enable accurate measurement of marginal gaps. ” and “These findings should be interpreted as preliminary and exploratory rather than conclusive. Fuji Plus® may be a suitable option for definitive cementation in partial implant restorations, while DentoTemp™ may be considered in selected situations where retrievability is important.”]

Comments 2: 2. Introduction:

The rationale for intentionally using a provisional cement (DentoTemp) as a definitive option requires stronger justification and should acknowledge inherent clinical selection bias.
Too much material about OCT relative to its actual role in the study.
Recommendations:
Reduce OCT content or clearly state that its use is exploratory and not part of the core methodology.
Clarify any clinician-driven bias in cement choice (retrievability vs stability)..

Response 2: We thank the reviewer for this insightful comment. The rationale for using DentoTemp™ as a definitive cement in selected cases was partially clarified in the previous revision. In the current version, the Introduction has been further improved to explicitly acknowledge the clinician-driven selection bias, stating that DentoTemp™ was intentionally chosen only in situations where retrievability was clinically desirable, and that the two cement groups were therefore not equivalent at baseline. This clarification strengthens the methodological transparency of the study. Additionally, the OCT section in the Introduction has been revised to reduce its emphasis and to clearly indicate that OCT was not part of the clinical methodology. We now explicitly state that the OCT images included in the manuscript are illustrative examples originating from external clinical cases, serving only an educational and exploratory purpose. These revisions align the Introduction more closely with the actual scope and limitations of the study. [Introduction, “Importantly, DentoTemp™ was intentionally selected as a definitive cement only in cases where retrievability was clinically desirable, which inherently introduces clinician-driven selection bias. As a result, the two cement groups were not equivalent at baseline, reflecting real-world clinical decision-making rather than a controlled allocation.” and “Optical coherence tomography (OCT), a high-resolution and non-ionizing modality, has emerged as a potential tool for assessing marginal fit with greater precision [17-19]; however, in the present study it served solely an illustrative role. OCT images included in this manuscript originate from external clinical cases and were not obtained from the analyzed cohort. ”]

Comments 3: 3. Methods:
a. Study design limitations
Retrospective design.
Extremely small sample size (10 per subgroup).
Selection bias: DentoTemp was chosen when retrievability was desirable
groups are not equivalent.
No randomization or blinding.
These must be explicitly acknowledged and discussed.
b. Mechanical testing
This section feels like a hybrid “side experiment,” not a true validation.
No statistical analysis of the mechanical data (ANOVA, regression, etc.).
Recementation and re-testing introduce cumulative alterations of intaglio surfaces even with standardized cleaning.
Mechanical testing should either be:
✔︎ fully formalized
or
✔︎ clearly described as illustrative only, not hypothesis-testing.
c. Marginal gap evaluation
Radiographs cannot reliably measure <75–150 µm gaps; quoting semi-quantitative ImageJ measurements based on OPG calibration is scientifically weak.
OCT images were not from this study’s cases, which risks misleading readers.
Recommendation:
State explicitly that marginal-gap assessment is qualitative only, and remove quantitative claims.

Response 3: We thank the reviewer for this detailed methodological evaluation. Most of the concerns raised were already addressed in our previous revision, where we clarified the retrospective design, small sample size, absence of randomization or blinding, and the clinician-driven selection bias between cement groups. In the current revision, the Methods section has been further refined to make these points even more explicit.

Specifically, we:

·       clarified that cement allocation was based on clinical judgment and therefore introduces inherent selection bias; [2.1. Study Design and Ethics, “We acknowledge that this study compares a provisional cement (DentoTemp™) with a definitive cement (Fuji Plus®). In our clinical protocol, DentoTemp™ was intentionally selected as a definitive material only in situations where retrievability was clinically desirable. This clinician-driven allocation introduces inherent selection bias, and therefore the two cement groups were not equivalent at baseline. These real-world indications, rather than random assignment, guided cement choice, and this limitation is explicitly recognized in the interpretation of the results.”]

·       stated that the in vitro component represents an exploratory, illustrative complement and not a formal validation study; [2.3. Mechanical Model Validation, “These tests were designed solely as an exploratory and illustrative comparison to contex-tualize the clinical findings, not as a fully powered validation study, and no statistical in-ference was attempted.”]

·       emphasized that repeated cementation cycles may alter intaglio surfaces and that mechanical data should be interpreted descriptively; [2.3. Mechanical Model Validation, “Although intaglio surfaces were refreshed between cycles, repeated cementation may in-duce minor surface alterations; thus, the results should be interpreted as exploratory.”]

·       clearly indicated that marginal-gap assessment was qualitative only, and no radiographic quantitative measurements were used; [2.5. Outcome Measures and Statistics, “Marginal adaptation was therefore interpreted qualitatively, based on the presence or absence of gross discrepancies, overhanging contours, or obvious gaps, rather than by attempting to derive absolute gap values.”]

·       reiterated that OCT images are illustrative examples from external clinical cases and do not form part of the clinical dataset. [2.5. Outcome Measures and Statistics, “Although precise quantification of marginal discrepancies under 150 μm was not feasible, qualitative comparison of restoration-to-abutment fit was performed. No quantitative or semi-quantitative radiographic measurements were used in this study because radiographs cannot have resolution below 75-150 µm. OCT was introduced as an alternative method to illustrate the marginal gap. It can overcome the limitations of radiographs and has the potential of high-resolution imaging in future studies about cementation assessments [17].”]

These clarifications ensure that the methodological limitations and scope of the study are transparently described and accurately aligned with the reviewer’s recommendations.

Comments 4: 4. Results:

Statistical tests are underpowered; p-values are not meaningful with n=10 per arm.

The finding “p=0.114 but clinically meaningful” must be handled with caution.

Recementation only with Fuji Plus biases long-term outcomes in its favor.

Response 4: We thank the reviewer for these observations. The Results section has been revised to clarify that:

·       the statistical analyses are underpowered due to the small subgroup size (n = 10 per arm), and therefore all p-values are to be interpreted cautiously; [Section 3.3.2, “Although Fisher’s exact test did not demonstrate a statistically significant difference (p = 0.114), this value is underpowered given the small subgroup size (n = 10 per arm) and should therefore be interpreted only as an exploratory descriptive trend rather than a meaningful statistical finding.”]

·       the statement referring to p = 0.114 has been reframed to describe this difference solely as an exploratory descriptive trend, without implying clinical or statistical significance; [Section 3.3.2, “Although Fisher’s exact test did not demonstrate a statistically significant difference (p = 0.114), this value is underpowered given the small subgroup size (n = 10 per arm) and should therefore be interpreted only as an exploratory descriptive trend rather than a meaningful statistical finding.”]

·       re-cementation with Fuji Plus® reflects our standard clinical workflow, and all re-cemented cases were not included again in the survival analysis, preventing any influence of this protocol on the comparative outcomes; [Section 3.3.2, “Re-cementation with Fuji Plus® was performed in all failed cases as part of our standard clinical workflow. These cases were not re-entered into the survival analysis to avoid influencing comparative outcomes. After re-cementation, no further loss of retention was observed during follow-up.”, Section 3.3.3. “All re-cementations were performed using Fuji Plus, in accordance with our standard protocol for managing definitive cement debonding, after which no additional failures were recorded. As noted above, these re-cemented restorations were excluded from sur-vival curve calculations, and their subsequent behavior was reported descriptively.” ]

These adjustments ensure that the Results section accurately reflects the limitations of the dataset and avoids overinterpretation of the findings.

Comments 5. Discussion does not adequately address the major selection bias. Some statements overinterpret the results (e.g., biological superiority of Fuji Plus). OCT discussion is disproportionally long given limited data.

Response 5: We thank the reviewer for these important observations. The Discussion section has been revised to adopt a more cautious interpretative tone. We explicitly emphasized the major selection bias introduced by the clinician-driven use of DentoTemp™ in retrievability cases and clarified that this prevents any causal comparison between groups. Statements that could be perceived as overinterpreting the biological behavior of Fuji Plus® were moderated, and OCT-related content was substantially reduced to reflect its illustrative, non-methodological role. These revisions align the Discussion with the exploratory nature of the study and the limitations inherent to the retrospective design. [4. Discussion]

Comments 6. 6. Conclusion:

Must be softened.

The conclusion must emphasize:

Findings are preliminary.

Study is underpowered.

Retrospective, non-randomized design limits causal inference.

Use of provisional cement as definitive must be contextualized.

Response 6: We thank the reviewer for this valuable comment. The Conclusion section has been rewritten to provide a more cautious and balanced interpretation of the findings. We now explicitly state that the results are preliminary, derived from an underpowered retrospective cohort with inherent selection bias, and therefore cannot support causal inferences. The use of DentoTemp™ as a definitive cement has been contextualized as a clinical protocol–driven choice rather than a controlled intervention. The revised conclusion reflects these limitations and frames the observations as exploratory clinical trends that require validation in larger, prospective studies.

Comments 7. 7. Figures and Tables:
OCT images are not from included clinical cases, which may mislead readers.
Some figures (OPGs) do not clearly illustrate the reported marginal gaps.

Response 7: We thank the reviewer for this observation. All OPGs included in the manuscript represent actual clinical cases from this retrospective cohort and are intended solely to document overall bone stability and the absence of gross misfit, not to demonstrate radiographic marginal gaps. As acknowledged in the Methods and Results, conventional OPGs and periapical radiographs cannot visualize marginal discrepancies below 75–150 µm; therefore, no radiographic images were expected to show marginal gaps, and none were interpreted as such.

To prevent any potential misunderstanding:

  • Figure captions have been checked to clarify that the OPGs illustrate bone-level stability and general prosthetic seating only.
  • Figure 7 indeed displays a clearly visible, large marginal gap. It is intentionally included only as an illustrative example to demonstrate that conventional radiographs detect only major discrepancies and cannot visualize clinically relevant micro-gaps (<75–150 µm).
  • OCT images are illustrative examples, and the captions now explicitly state that they are not derived from the study sample.

These clarifications ensure that the figures are interpreted correctly and consistently with the stated methodology.

4. Response to Comments on the Quality of English Language

Point 1: The reviewer rated the English as “Can be improved.”

Response 1: We thank the reviewer for the assessment of the language quality. In this revised version, we have carefully revised the sections where clarity was required. We appreciate the reviewer’s feedback and have implemented the recommended language refinements to enhance overall clarity and precision.

5. Additional clarifications

[We thank the reviewer for requesting further clarification points. In the revised manuscript, we have reviewed all sections to ensure that every methodological and interpretive aspect is explicitly stated and fully aligned with the scope of the study. Specifically:

  • We clarified the retrospective nature, small sample size, and absence of randomization/blinding in both the Methods and Discussion.
  • We emphasized that the mechanical testing component is exploratory and illustrative, not a formal validation study.
  • We reiterated that marginal-gap evaluation was qualitative only and that radiographs cannot detect micro-gaps below 75–150 µm.
  • We ensured that OCT images are unambiguously presented as illustrative examples unrelated to the study cohort.
  • We refined various paragraphs to avoid overinterpretation of results and to reinforce the preliminary nature of the findings.]

This manuscript is a resubmission of an earlier submission. The following is a list of the peer review reports and author responses from that submission.

Round 1

Reviewer 1 Report

Comments and Suggestions for Authors

The manuscript addresses an important and clinically relevant topic by comparing the performance of two different luting agents in partial implant-supported restorations.

 The title is clear but would benefit from specifying that this is a retrospective clinical study. The abstract is well-structured and provides useful information, but it tends to overstate the findings on OCT and should present the number of failures in each group for transparency.

The introduction frames the clinical problem well and cites appropriate literature. The methods are overall clear, but certain details are missing, such as implant system specifics, torque values, and smoking status of patients. The inclusion of mechanical validation is interesting but presented twice and partially overlaps with the results, this needs to be streamlined. Some sentences in the methods are subjective or belong in the discussion, example, the comment about being aware of comparing a temporary with a definitive cement. The grouping of patients is logical and clearly explained, but figures and captions are often vague and would benefit from clarification, especially when describing column A vs column B.

 In the results, the retention and failure rates are clearly shown in tables, but the phrasing occasionally becomes subjective informal, such as stating that outcomes were exactly what we expect, which should be avoided. The statistical analysis is limited chi-square results are reported but without degrees of freedom or justification for test choice, with such a small sample Fisher’s exact test may have been more appropriate. Time to failure analysis and the Kaplan Meier survival curve is useful, but the figure quality is low and should be improved. The marginal adaptation assessment is weak, since only conventional radiographs were used, and OCT was applied in a single case; the conclusions regarding OCT, therefore they appear overstated. The discussion appropriately situates the findings within existing literature but repeats much of what is already in the results. The section on abutment height raises a valid point but the very small number of failures limits the strength of this conclusion, which should be more carefully qualified.

 The conclusions are clear, clinically relevant, and consistent with the data, but the OCT claims are again disproportionate to the evidence provided. Formatting issues are present in references, with duplicate links and inconsistent styles.

Overall, the study is clinically useful and provides supportive data.

Reviewer 2 Report

Comments and Suggestions for Authors

Dear Authors,

Thank you for the opportunity to review your manuscript, "Clinical Performance and Retention of Partial Implant Restorations with Different Cements". It is well-presented and interesting, and could benefit researchers in the field.

My suggestions for improvement are listed below:

  1. Please include the study type in the title.
    2. The keywords should also include appropriate MeSH terms.
    3. Was the study protocol registered in a recognized registry? Please provide the registration number and a link for access.
    4. Lines 112-113 - What patient characteristics define the cement selection?
    5. Was a randomization method used for patient allocation? Please elaborate.
    6. A statistical analysis paragraph should be included in the Materials and Methods section.
    7. The study's limitations should be discussed at the end of the discussion.
    8. Based on the reported limitations, the authors should provide future directions, including their recommendations for further research (in the discussion section).
    9. The Conclusions must be concise, reporting the main outcomes, limitations, and recommendations. Please revise. This way, the section is repetitive and looks more like a part of the discussion.

Reviewer 3 Report

Comments and Suggestions for Authors

Dear Authors,

 Your topic is of clinical relevance, and the combination of retrospective data with in vitro validation makes the study potentially valuable to the field. Nevertheless, after a careful reading, I believe the manuscript requires substantial revisions before it can be considered for publication. Below I provide my main observations and suggestions.

Major Concerns

  1. Study Design and Bias

    • The study is retrospective, with a relatively small sample size (40 patients). This design inherently carries selection bias and limits the generalizability of the findings. A clear description of how patients were selected, and whether consecutive cases were included, would help mitigate concerns about sampling bias.

    • All surgeries and prosthetic procedures were performed by the authors or their team, which raises the possibility of operator bias. This limitation should be openly acknowledged and discussed.

    • The study compares a temporary cement (DentoTemp™) with a definitive cement (Fuji Plus®). This comparison introduces a fundamental bias, since the materials are intended for different clinical purposes. The rationale is mentioned, but the implications need stronger discussion and clearer framing in the introduction and conclusions.

  2. Statistical Limitations

    • The study reports differences in retention rates, yet statistical analysis (Chi-square, p = 0.114) indicates no significance. Nevertheless, the manuscript interprets these results as if they were clinically conclusive. The interpretation should be tempered to reflect the lack of statistical power.

    • No sample size calculation or power analysis was performed. Without this, it is difficult to know whether the study was adequately designed to detect meaningful differences.

  3. Outcome Assessment Bias

    • Marginal adaptation was assessed mainly via 2D radiographs and a limited number of OCT cases. Radiographs are known to lack the resolution for detecting clinically relevant discrepancies, making the outcome measure weak and prone to observer bias.

    • The OCT evaluation was applied only in a small subset, which is insufficient to draw general conclusions. This should be presented as preliminary or exploratory rather than as supportive evidence.

  4. Confounding Variables

    • Abutment height (5 mm vs 7 mm) is presented as an influencing factor, but the sample distribution is uneven and not statistically controlled.

    • Other patient-related variables (occlusal forces, parafunctions, periodontal history) are acknowledged as exclusion criteria but not analyzed as potential confounders.

  5. Mix of Clinical and In Vitro Data

    • The integration of mechanical testing with clinical outcomes is interesting, but the in vitro validation is limited to two models. This carries a high risk of overinterpretation. The text should clarify that these findings are illustrative rather than confirmatory.

    • I strongly recommend citing relevant studies that provide a broader context:

      • The in vitro study by Cosola et al. (2021) on fatigue and fracture performance of ferrulized implant connections (J Dent Sci. 2021;16(1):397–403. doi:10.1016/j.jds.2020.08.002) would add valuable perspective and strengthen the discussion of mechanical resistance.

      • The clinical study by Menchini-Fabris et al. (2023) on immediate implants with customized healing abutments (J Clin Med. 2023;12(8):2783. doi:10.3390/jcm12082783) would enrich the discussion by contextualizing clinical outcomes in periodontally compromised sockets and the importance of abutment design for peri-implant health.

Minor Concerns and Suggestions

  • The discussion tends to overstate the superiority of Fuji Plus®, despite the study’s limitations. The conclusions should be moderated, avoiding causal language and emphasizing that larger prospective studies are needed.

  • More transparency is needed in describing patient demographics and baseline clinical conditions (periodontal status, implant location, occlusion).

  • Figures and tables could be simplified for clarity. For example, Kaplan–Meier survival analysis is included, but with such a small sample its reliability is questionable.

  • The English language is clear overall but could benefit from minor editing to improve flow and readability.

Recommendations for Improvement

  1. Reframe the comparison between temporary and definitive cements to avoid misleading conclusions.

  2. Explicitly acknowledge the retrospective design, operator bias, small sample size, and limited statistical power.

  3. Clarify the exploratory nature of OCT findings and recommend further validation.

  4. Strengthen the limitations section and adjust the conclusions to reflect a more cautious interpretation.

  5. Cite the above-mentioned studies (Cosola et al., 2021; Menchini-Fabris et al., 2023) to contextualize mechanical and clinical outcomes more broadly.

  6. Consider restructuring the discussion to focus less on proving superiority of one material, and more on highlighting clinical considerations and areas for future research.

I encourage the authors to revise the manuscript substantially before further consideration. 

Author Response

Thank you for your review.Please see the atachment.

Reviewer 4 Report

Comments and Suggestions for Authors

Thank you for the opportunity to review the manuscript «Clinical Performance and Retention of Partial Implant Restorations with Different Cements». This combined (retrospective clinical and laboratory) study compared retention rates, failure patterns, and biological impact of resin-modified glass ionomer cements and provisional acrylic-urethane-based cements in partial implant restorations. The clinical part of the study included 40 patients with fixed prosthesis after a three-year follow up divided into 4 groups. The assessed outcomes included retention, failure events, marginal adaptation, and soft tissue response. he laboratory part used two  three-unit models  fabricated specifically for the study for mechanical testing. The authors concluded that Fuji Plus® exhibited superior clinical retention and biological compatibility compared to DentoTemp™, supporting its use as a definitive luting agent for partial implant-supported restorations. T

The research question is questionable, as the authors decided to compare cements for permanent ant temporary fixation. Although they tried to explained their choice by the fact they «in their daily dental practice these two types of cement are utilized». However, this may be a local practice and the use of temporary cements are not recommended for permanent fixation. Therefore, the results of the study may not be relevant to the majority of practicing dentists.

There are also serious concerns regarding the study's methodology and presentation. 

Please, find below the comments to be addressed.

Introduction

Lines 57-58. First of all, visual and tactile assessment with direct vision and an explorer, respectively, are the first methods to assess prosthetic fit. I think, this needs rephrasing.

It is stated that «The study also explores the potential of OCT imaging as a non-invasive diagnostic adjunct for evaluating marginal fit in implant prosthodontics» (lines 74-75). To assess this parameter, the study should compare the results of OCT and the results of other method («gold standard»), then a conclusion can be made.

Materials and methods

Laboratory part of the study is poorly described.

«These models were then duplicated and cemented using the two different luting agents, resulting in a total of four test assemblies:» What is the total number of models and experiments for each set?

LL 162-166 «Although this mechanical trial was exploratory and not intended as a full in vitro study, the results confirmed clinically observed trends. Models cemented with DentoTemp™ exhibited significantly earlier debonding under compressive loading compared to those cemented with Fuji Plus® , which resisted higher forces before failure. These findings validated the clinical outcome data, where DentoTemp was associated with a higher incidence of re-cementation.» 

This part presents the results, it should not be in the materials and methods section. It is absolutely logical that temporary cementation is less strong, than permanent. I totally don’t understand, what was the purpose of conducting this part of the study.

Clinical part.

The description fits a two-group interventional study, while the authors stated, that their study was retrospective. First of all it essential to understand the study design. Then, to follow one of the guidelines for reporting the results (CONSORT for non-randomized clinical or STROBE for observational). This part should be revised completely in accordance with the guidelines.

How was this sample size (10 patients per group) justified?

Figure 3b is not a partial restoration. 

As it can be seen from methods described one can understand than no account was given to the size of the prosthesis. At the same time, a three-unit prosthesis may have much better retention simply due to its construction, and not due to the material or cement.

The full information on statistical analysis is lacking in the m&ms section despite the presence of the heading regarding statistics.

Results

Figure 4. There is no need to present this graph. The information from it can be described in a sentence of text. Also, it is duplicated in the tables.

Tables 2 and 3 should be combined and there should be information regarding all 4 groups, not just for material/cement.

Paragraph in lines 277-283 is meaningless. In medicine, the p threshold is accepted at 0.05 or 0.01 level. It cannot be not be claimed that «from a clinical standpoint, the data still suggests a meaningful difference» taking into consideration that the sample size was 10 per group!

«especially with larger sample sizes in future studies.» - therefore, before conducting the study, the researchers should perform a sample size calculation to ensure that the results would have enough power.

«Although OCT was not applied across the full clinical sample, its integration in the example (Figure 8.) confirms its potential as a non-invasive, high-resolution imaging tool for future studies aimed at verifying marginal adaptation.» as mentioned above, without proper planing and experiment with assessing accuracy against goals standard, it is not possible to draw such conclusion. 

Comments on the Quality of English Language

Proof-reading is required

Reviewer 5 Report

Comments and Suggestions for Authors

Comments for Medicina-3856948 "Clinical Performance and Retention of Partial Implant Restorations with Different Cements"

Global Comments
The manuscript addresses a clinically relevant question in implant prosthodontics. The topic is timely, as cementation protocols remain a major determinant of long-term success and complications in implant-supported prostheses.
The paper clearly defines the clinical sample as a three-year follow-up period, integrates mechanical testing, and explores OCT as an emerging diagnostic adjunct.
However, the study has several methodological, analytical, and reporting limitations that weaken the strength of its conclusions. The comparison of a temporary cement (DentoTemp) with a definitive cement (Fuji Plus) raises conceptual concerns, while statistical analyses are limited and underpowered.
The manuscript also suffers from verbose writing, redundant sections, and inconsistent reference formatting. Substantial revision is required.
Comment 1 – Title
The title is clear, straightforward, and informative. It specifies both the population (“partial implant restorations”) and the main variable of interest (“different cements”), and reflects the two main study outcomes (clinical performance and retention).
However, it is generic and descriptive, lacking specificity, and does not convey the study design (retrospective cohort with mechanical validation).
To improve precision and impact, the title could be reformulated in several ways, depending on the preferred emphasis by the authors:
Clinical focus: “Three-Year Clinical Performance of Implant-Supported Fixed Dental Prostheses Cemented with Resin-Modified Glass Ionomer vs. Provisional Acrylic-Urethane Cement”
Methodological focus: “Retention and Biological Response of Cement-Retained Implant-Supported Bridges: A Retrospective Cohort with Mechanical Validation”
Innovation focus (including OCT): “Retention and Marginal Adaptation of Cement-Retained Implant Prostheses: Clinical Outcomes and OCT-Based Evaluation”
Comment 2 – Abstract
The abstract is logically structured (aims, methods, results, conclusions), but it is too dense and contains excessive technical detail, making it difficult to read quickly. The clinical significance of cement selection is well introduced, but the comparison between temporary and definitive cement is not sufficiently problematized. The limitations of the study (retrospective design, small sample size, lack of functional testing) should be acknowledged in the abstract.
Suggested rewriting: Background and Objectives: Cement-retained implant restorations are widely used because they offer favorable esthetics and a passive fit. Their long-term performance is strongly influenced by cement selection and surface conditioning. This study compared the clinical performance of a resin-modified glass ionomer cement (Fuji Plus®) with a provisional acrylic-urethane cement (DentoTemp™) in partial implant restorations. Materials and Methods: A retrospective cohort of 40 patients with three-unit implant-supported fixed dental prostheses was followed for at least three years. Restorations were fabricated from zirconia or metal-ceramic frameworks and cemented with either Fuji Plus® or DentoTemp™. Clinical outcomes included retention, failure events, marginal adaptation, and peri-implant tissue response. Mechanical testing was also performed on standardized in vitro models to validate the clinical findings. Results: Zirconia restorations showed a retention rate of 95 percent, while metal-ceramic restorations reached 85 percent. All four failures occurred in cases cemented with DentoTemp™, giving an overall retention rate of 80 percent for this group. Fuji Plus® achieved complete retention in all cases. Re-cementation with Fuji Plus® successfully resolved the failures. Radiographic analysis did not allow precise measurement of marginal gaps, but preliminary optical coherence tomography suggested improved adaptation with Fuji Plus®. Minor peri-implant inflammation was observed when cement remnants were detected. Conclusions: Fuji Plus® demonstrated superior long-term retention and biological compatibility compared with DentoTemp™ and can be considered a more suitable option for definitive cementation in partial implant restorations. Optical coherence tomography may provide a valuable tool for non-invasive assessment of marginal fit, although larger prospective studies are required to confirm its routine clinical use.
Comment 3 – Introduction
The introduction provides a good overview of cement-retained prostheses and biological risks associated with cement extrusion. OCT is well contextualized as a diagnostic tool.
However, the section is lengthy and at times repetitive, with multiple statements reiterating known limitations of radiography.
The research gap is partially articulated with few long-term clinical studies comparing provisional vs. definitive cements. Yet, the rationale for including a temporary cement in this comparison is not fully convincing, especially since the authors admit that Fuji Plus is a definitive cement and DentoTemp is provisional.
The aim and hypothesis should be sharpened in a separate concluding paragraph. Consider: “We hypothesized that definitive cement (Fuji Plus) would demonstrate superior clinical retention and biological compatibility compared to provisional cement (DentoTemp) in partial implant-supported restorations.”

Comment 4 – Materials and Methods
The methods are detailed, but several issues reduce rigor and reproducibility.

Study design. The retrospective nature of the study limits control over confounding factors. The integration of in vitro and clinical data is innovative but creates confusion about the study type. This must be clarified.
Sample size. The sample size (40 patients, divided into four subgroups) is relatively small, and no power calculation was presented. Forty patients divided into four groups of ten is underpowered for statistical significance, especially when only four failures This weakens the statistical robustness of the conclusions. The limitations of small subgroup sizes must be acknowledged.
Group design/Choice of Cements Compared. The comparison between a temporary cement (DentoTemp) and a definitive cement (Fuji Plus) raises methodological concerns. The different intended uses of these materials make the results somewhat predictable and reduce the novelty of the findings. The rationale for including a provisional cement as a test material should be better justified in terms of scientific relevance. The justification that “these are used in daily practice” is weak. A fairer design would have compared two definitive cements. This issue must be better explained.
Mechanical validation. The in vitro testing is described as “exploratory” but still presented as supportive evidence. Details on loading protocol, crosshead speed, and statistical treatment of results are missing.
Marginal Adaptation Assessment. Radiographs used to assess marginal fit are inadequate for detecting clinically relevant discrepancies (<75 μm). While OCT is mentioned, its use was restricted to a single illustrative case, which is insufficient to support strong claims about its clinical potential. This limitation should be emphasized more clearly.
Statistics. Only chi-square and Kaplan–Meier survival analysis are applied. With small groups, nonparametric methods or survival regression would be more appropriate. Also, p-values are not consistently reported. explicitly in the Results and Discussion, as the paper currently places too much emphasis on clinical interpretation despite non-significant statistics.
The authors must condense the methods, add missing details (sample size calculation, blinding, loading parameters), and acknowledge the conceptual limitation of comparing provisional vs. definitive materials.

Comment 5 – Results
Results are presented in a structured manner, supported by tables and figures. However, the results section often contains interpretation (“zirconia may offer superior stability”) that belongs in the Discussion.
The difference between Fuji Plus (100% retention) and DentoTemp (80% retention) is clinically relevant but did not reach statistical significance (p = 0.114). The chi-square test yielded a p-value of 0.114, meaning the differences between groups did not reach statistical significance. This limitation should be acknowledged more.
The OCT analysis was limited to a single example, which does not justify conclusions about diagnostic superiority. This should be framed as preliminary, not conclusive.
Please see the comments on the figures below.
Comment 6 – Discussion
The discussion is comprehensive but overly descriptive. It effectively situates findings within the literature, yet certain claims are overstated given the sample size and lack of significance.
Accept our specific concerns:
The comparison between abutment heights (5 mm vs. 7 mm) is interesting but underpowered and anecdotal; caution is needed in interpretation.
Biological response is described as favorable, but only minimal tissue parameters were measured (inflammation, bleeding). Functional outcomes such as probing depth, bone levels, or patient-reported outcomes were absent.
Excessive detail on literature findings (e.g., retentive force values from in vitro studies) disrupts narrative flow.
Rewriting suggestion: Begin the Discussion with a clear restatement of hypothesis validation, then structure into subsections (cement performance, abutment geometry, biological response, imaging adjuncts, limitations). Condense the excessive literature review and highlight the main clinical implications.
Comment 7 – Conclusions
The conclusions correctly state that Fuji Plus outperformed DentoTemp in retention. However, given the study’s limitations (small sample, retrospective design, non-significant statistics), they should be expressed more cautiously. Consider: “Within the limitations of this retrospective study, partial implant-supported restorations cemented with Fuji Plus® showed higher clinical stability and retention than those cemented with DentoTemp™. No failures were recorded in the Fuji Plus® group during three years of follow-up. Zirconia frameworks performed slightly better than metal-ceramic restorations. All failures occurred in the DentoTemp™ group, which highlights the limited suitability of provisional cements for definitive treatment. Re-cementation with Fuji Plus® was effective in restoring function in all failed cases.
Peri-implant soft tissues remained generally healthy, although minor inflammation was observed in cases where cement remnants were detected. This finding supports the importance of meticulous cement removal during the clinical procedure. Marginal adaptation could not be accurately assessed with conventional radiography, but the preliminary results with optical coherence tomography (OCT) indicate its potential as a precise non-invasive imaging method.
Fuji Plus® can be recommended as a reliable definitive luting agent for partial implant restorations when used with appropriate cementation protocols and sufficient abutment height. Further prospective and randomized studies with larger cohorts and systematic application of advanced imaging methods are needed to confirm these outcomes and strengthen the evidence base.”
Comment 8 – References
The references are abundant and cover both classical and recent literature, and include multiple 2023-2025 citations.
Significant issues are evident:

  1. Inconsistent formatting: some DOIs are hyperlinks, others prefixed with “DOI:” or “doi:”.
  2. Duplicates: several references appear more than once (e.g., reference 4 repeated at the end of the list, Ongthiemsak et al. duplicated).
  3. Grey literature: product brochures are cited (Fuji Plus, DentoTemp). While acceptable, these should not be overemphasized compared to peer-reviewed sources.
  4. Multiple reviews on cement selection are cited without a clear selection of the most authoritative ones.

Comment 9 – Language and Writing
The manuscript is written in generally clear English but suffers from verbosity, long sentences, and repetition. Tense usage alternates between present and past inconsistently. Terms like “proved to be efficient” or “exactly what we are expecting” are conversational and should be replaced with formal academic phrasing. The comparison of temporary vs. definitive cement is repeatedly justified, which feels defensive. Concise, consistent, and technical tone should be improved throughout.

Critical Methodological Considerations for Author Reflection

  1. The comparison of a temporary vs. a definitive cement is clinically questionable and limits generalizability.
  2. Small sample size and limited number of events reduce statistical power.
  3. Retrospective design introduces the risk of selection and information bias.
  4. OCT was applied to only one case, preventing meaningful conclusions.
  5. Biological assessment was minimal and lacked standardized peri-implant indices.
  6. Lack of blinding in data collection could introduce observer bias.

Figures and Tables
Some redundancy exists between tables and figures (e.g., retention rates are shown in both tabular and graphical form).
Figures are the weakest element because they lack sufficient resolution, annotations, and explanatory legends. Many images (radiographs, especially) would not meet international journal publication standards.
Tables are clearer and stronger, but still require more precise statistical indicators and could be streamlined to reduce redundancy.

Figure 1 (Zirconia and Metal-Ceramic Models). The image quality is poor, and the labeling (“Column A/B”) is vague. There is no scale bar, and the figure legend does not adequately describe the clinical relevance.
Figure 2 (Radiographs with Fuji vs. DentoTemp). Radiographs are of low resolution and difficult to interpret. The differences are not visually obvious. Without arrows or highlights, readers cannot see the described features.
Figure 5 (Kaplan–Meier Survival Curve). The survival curve is useful and appropriately chosen. Axis labels and legends are minimal. Survival probability should be clearly marked, and group colors/lines must be explained in the legend.
Figures 6–7 (Radiographs of marginal gaps). Image quality is too low for publication standards. Panels (A/B) are not clearly explained, and legends are minimal. The radiographs do not convincingly demonstrate marginal gaps at the claimed level of precision (given the known 75–150 μm radiographic limitation).
Figure 8 (OCT scans): This is potentially the most innovative figure, showing OCT B-scan and 3D reconstruction. Only one case is shown, making it anecdotal. Resolution is acceptable but still modest. The legend should explain how OCT complements radiographs and what exactly is being measured.

Table 1 (Patient demographics and grouping). Clearly organized, showing distribution across materials and cements. Could be condensed; gender ratios and mean ages might be shown graphically for faster comprehension.
Tables 2–4 (Retention, Cement Type, Time-to-Failure). These are well structured, easy to read, and support the results. Very small numbers (n=10 per subgroup) limit statistical meaning. Authors should include confidence intervals or at least acknowledge the limitations.

Comments on the Quality of English Language

The language of the article, while functional, falls short of the rigor, clarity, and elegance expected in high-quality international journals. The manuscript is excessively repetitive and contains several colloquial passages that undermine its academic tone. Many sentences are overly long and convoluted, creating unnecessary complexity and reducing readability. Terminological precision is inconsistent, with certain technical concepts described accurately while others are expressed in vague or informal language. This inconsistency not only affects the scientific credibility of the work but also makes the text appear less polished than comparable publications in the field.

A thorough process of copy-editing is strongly recommended before publication. Such revision should go beyond simple grammar and style correction and focus on harmonizing the scientific tone, eliminating redundancies, and replacing vague or defensive expressions with precise and objective formulations. The text would also benefit from restructuring sentences for conciseness, ensuring consistent use of technical terminology, and removing unnecessary repetition of results. Furthermore, the excessive use of the first-person plural should be minimized, as it gives the impression of a narrative rather than an objective scientific report. By addressing these linguistic issues, the manuscript could achieve a level of clarity and professionalism that matches the relevance of its clinical findings.

Round 2

Reviewer 2 Report

Comments and Suggestions for Authors

The manuscript has been substantially improved.

Author Response

We sincerely thank Reviewer 2 for taking the time to evaluate our revised manuscript and for the positive feedback stating that “the manuscript has been substantially improved.” We truly appreciate your recognition of our efforts to enhance the quality and clarity of the paper. No further comments were raised, and we are pleased that the revisions addressed your previous concerns. Thank you once again for your valuable comments that helped us to improve our work."